# Insight into the Structure and Redox Chemistry of [Carbonatotetraamminecobalt(III)] Permanganate and Its Monohydrate as Co-Mn-Oxide Catalyst Precursors of the Fischer-Tropsch Synthesis

Kende Attila Béres [1,2], Zsolt Dürvanger [3,4], Zoltán Homonnay [5], Laura Bereczki [1,6], Berta Barta Holló [7], Attila Farkas [8], Vladimir M. Petruševski [9] and László Kótai [1,10,*]

1. Institute of Materials and Environmental Chemistry, HUN-REN Research Centre for Natural Sciences, H-1117 Budapest, Hungary; beres.kende.attila@ttk.hu (K.A.B.); nagyne.bereczki.laura@ttk.hu (L.B.)
2. György Hevesy PhD School of Chemistry, ELTE Eötvös Loránd University, H-1053 Budapest, Hungary
3. Structural Chemistry and Biology Laboratory, Institute of Chemistry, ELTE Eötvös Loránd University, H-1117 Budapest, Hungary; zsolt.durvanger@ttk.elte.hu
4. ELKH-ELTE Protein Modelling Research Group, H-1117 Budapest, Hungary
5. Institute of Chemistry, ELTE Eötvös Loránd University, H-1053 Budapest, Hungary; homonnay.zoltan@ttk.elte.hu
6. Centre for Structural Science, HUN-REN Research Centre for Natural Sciences, H-1117 Budapest, Hungary
7. Department of Chemistry, Biochemistry and Environmental Protection, Faculty of Sciences, University of Novi Sad, SRB-21000 Novi Sad, Serbia; hberta@uns.ac.rs
8. Department of Organic Chemistry and Technology, Faculty of Chemical Technology and Biotechnology, Budapest University of Technology and Economics, H-1117 Budapest, Hungary; farkas.attila@vbk.bme.hu
9. Institute of Chemistry, Faculty of Natural Sciences and Mathematics, Ss. Cyril and Methodius University, MK-1000 Skopje, North Macedonia; vladimirpetrusevski@yahoo.com
10. Deuton-X Ltd., H-2030 Érd, Hungary
* Correspondence: kotai.laszlo@ttk.hu

**Abstract:** [Carbonatotetraamminecobalt(III)] permanganate monohydrate was synthesized first in the metathesis reaction of $[Co(NH_3)_4CO_3]NO_3$ and $NaMnO_4$ in aqueous solution. Its thermal dehydration at 100 °C resulted in phase-pure $[Co(NH_3)_4CO_3]MnO_4$ (compound **1**). Compounds **1** and **2** (i.e., the hydrated form) were studied with IR, far-IR, and low-temperature Raman spectroscopies, and their vibrational modes were assigned. The lattice parameters were determined by powder X-ray diffraction (PXRD) and single crystal X-ray diffraction (SXRD) methods for the triclinic and orthorhombic compounds **1** and **2**, respectively. The detailed structure of compound **2** was determined, and the role of hydrogen bonds in the structural motifs was clarified. UV studies on compounds **1** and **2** showed the distortion of the octahedral geometry of the complex cation during dehydration because of the partial loss of the hydrogen bonds between the crystal water and the ligands of the complex cation. The thermal decomposition consists of a solid phase quasi-intramolecular redox reaction between the ammonia ligands and permanganate anions with the formation of ammonia oxidation products ($H_2O$, $NO$, $N_2O$, and $CO_2$). The solid phase reaction product is amorphous cobalt manganese oxide containing ammonium, carbonate (and nitrate) anions. The temperature-controlled thermal decomposition of compound **2** in toluene at 110 °C showed that one of the decomposition intermediates is ammonium nitrate. The decomposition intermediates are transformed into $Co_{1.5}Mn_{1.5}O_4$ spinel with $MnCo_2O_4$ structure upon further heating. Solid compound **2** gave the spinel at 500 °C both in an inert and air atmosphere, whereas the sample pre-treated in toluene at 110 °C without and with the removal of ammonium nitrate by aqueous washing, gave the spinel already at 300 and 400 °C, respectively. The molten $NH_4NO_3$ is a medium to start spinel crystallization, but its decomposition stops further crystal growth of the spinel phase. By this procedure, the particle size of the spinel product as low as ~4.0 nm could be achieved for the treatments at 300 and 400 °C, and it increased only to 5.7 nm at 500 °C. The nano-sized mixed cobalt manganese oxides are potential candidates as Fischer-Tropsch catalysts.

**Keywords:** redox reaction; ammine; carbonate; permanganate; hydrogen bond; crystal structure; vibrational spectroscopy; spinel; Fischer-Tropsch; thermal analysis

## 1. Introduction

An easy and convenient reaction route for the preparation of the simple and mixed nanosized transition metal oxides [1–18]—especially for the synthesis of catalytically active spinel oxides [19–30]—is the thermal decomposition of complexes having reducing ligands and oxidizing anions [31–38]. These processes consisted of a solid-phase quasi-intramolecular redox reaction between an oxidizing anion and a reducing ligand. This reaction route led to various cobalt manganese spinels, which were found to be active catalysts for various industrially important processes. The appropriate selection of the ammine-coordinated cobalt and permanganate-containing precursors can result in the formation of $Co_xMn_{2-x}O_4$ type catalysts with Co to Mn ratio 1:1–3 with exact compositions and properties depending on the preparation conditions as well [4,27,39–42]. With the use of [carbonatotetraamminecobalt(III)] permanganate (compound **1**, $Co(NH_3)_4CO_3]MnO_4$, with Co to Mn stoichiometry 1:1, Mansouri et al. prepared an excellent Fischer-Tropsch catalyst characterized with $CoMn_2O_4$ composition [4]. However, the decomposition of the [carbonatotetraamminecobalt(III)] permanganate (compound **1**) might result in $CoMn_2O_4$ formation only if at least one Co-rich (or pure) Co-compound is also formed together with it. Mansuri [4], however, did not mention the formation of any accompanying Co-compound. The studies of the supposed $CoMn_2O_4$ phase were performed by PXRD) [4,27], but both $Co_3O_4$ ($Co^{II}Co^{III}_2O_4$) and $Co^{II}Co^{III}_{0.5}Mn_{1.5}O_4$ ($Co_{1.5}Mn_{1.5}O_4$) are expected to give similar PXRD patterns as $CoMn_2O_4$ [42]. Therefore, we prepared phase-pure $[Co(NH_3)_4CO_3]MnO_4$ (compound **1**) and its monohydrate $[Co(NH_3)_4CO_3]MnO_4·H_2O$ (compound **2**) and decomposed them with isothermal heating to study the phase and composition changes. The compound prepared according to the literature process and expected to be $[Co(NH_3)_4CO_3]MnO_4$ proved to be its monohydrate, and we attempted to prepare the anhydrous salt as well. The structure of the monohydrate was elucidated with the SXRD method, and the spectroscopic and thermal properties of compounds **1** and **2** were studied and discussed in detail. The hydrogen-bond network in compound 1 and the perchlorate analog of the anhydrous salt, compound **1**-ClO$_4$, have been compared with the use of the Hirshfeld surface analysis method.

The compounds synthesized and evaluated, or their data compared with the title compounds parameters are given in Table 1.

**Table 1.** The compounds synthesized and used in the evaluation of the measured data.

| Compound | Label | Ref. |
|:---:|:---:|:---:|
| $[Co(NH_3)_4CO_3]MnO_4$ | **1** | [4,27] |
| $[Co(NH_3)_4CO_3]ClO_4$ | **1-ClO$_4$** | [43] |
| $[Co(NH_3)_4CO_3]MnO_4·H_2O$ | **2** | New compound |
| $[Co(NH_3)_4CO_3]NO_3$ | **3** | [44] |
| $[Co(NH_3)_4CO_3]NO_3·0.5H_2O$ | **4** | [45,46] |
| $[Co(NH_3)_4CO_3]_2SO_4·3H_2O$ | **5** | [47,48] |

## 2. Results and Discussion

### 2.1. Preparation and Properties of Compounds **1** and **2**

The only reaction route that was described to prepare compound **1** was the reaction of aq. [carbonatotetraamminecobalt(III)] nitrate solution and solid potassium permanganate with several minutes of stirring [4]. We have studied the reaction product with PXRD and found that the product is a multiphase system (Figure S1) because the amount of the water

used was not enough to dissolve the starting Co-complex (compound **3**), $KMnO_4$ was added as solid, and the sparingly soluble compound **1** can prevent the complete dissolution of $KMnO_4$ and compound **3** during the undefined reaction time. Accordingly, the Co to Mn ratio in the product mixture is not surely 1:1 as in the pure compound **1**. We repeated this synthetic route with the use of the extremely well-soluble [49] sodium permanganate solution when instead of compound **1**, its monohydrate, $[Co(NH_3)_4CO_3]MnO_4·H_2O$ (compound **2**) formed as purple needle-like crystals with 88.4% yield. The pH of the saturated aqueous solution of compound **2** in water is 5.33, and it is not soluble in ethanol and decomposes in diethyl ether and chloroform.

We tested numerous other preparation methods to isolate compounds **1** and **2**, known for the synthesis of permanganate salts [49,50]. However, all of these efforts completely failed due to the strong oxidative effect of high valence manganese species towards ammonia in the low-pH reaction media or due to the co-precipitation of the sparingly soluble compounds **1** or **2** ($s$ = 0.16 g/L and 0.15 g/L, respectively, in water at room temperature) with insoluble by-products as $BaSO_4$.

Therefore, we tried to remove the water content of compound **2** with careful isothermal heating of compound **1** at 100 °C because the [carbonatotetraamminecobalt(III)] cation core is stable at this temperature [47,48]. The isotherm heating experiments of compound **2** resulted in a decrease in the water content, keeping the purple color of compound **2**. The IR spectrum of the partially and completely dehydrated products (1 and 4 h heating at 100 °C, respectively) confirmed the presence of permanganate ions (Figure S2). It showed the decrease/disappearance of the intense bands belonging to symmetric and antisymmetric O-H stretching modes at ~3500 $cm^{-1}$, confirming the dehydration process (Figure S2). The PXRD of the partially and completely dehydrated products shows that the long-range ordering in the crystal structure decreases during the heating with increasing heating time (Figure S3). Furthermore, the formed anhydrous $[Co(NH_3)_4CO_3]MnO_4$ (Compound **1**) is not isomorphic with the analogous perchlorate, $[Co(NH_3)_4CO_3]ClO_4$ (Compound **1-ClO₄**) [43,51] (Figure S4).

Although Mansouri et al. studied some properties of compound **1** [4], the chemical nature of his sample, believed to be compound **1**, is questionable. First of all, the synthesis method (insufficient amount of water to dissolve the starting [tetraamminecobalt(III)] nitrate) and using solid $KMnO_4$ with undefined stirring time does not allow to reproduce their experiments in detail. The heat treatment of their reaction products resulted in a sample defined as $CoMn_2O_4$. However, the Co to Mn ratio in compounds **1** and **2** is 1:1, and $CoMn_2O_4$ with 1:2 Co:Mn stoichiometry cannot be formed. The X-ray diffraction (XRD) analysis of $(Co,Mn)^{II}(Co,Mn)^{III}_2O_4$ compounds [48] shows that their spinel structure gave similar patterns as $CoMn_2O_4$, so the identity of the reaction product with Co to Mn stoichiometry 1:2 is not confirmed. Even if we suppose that the stoichiometry of the reaction product is really 1:1, and the main component is compound **1** (the IR spectrum of their sample showed that there is no substantial amount of water in the sample), the TG curve shows a mass decrease supposed to be due to physisorbed/crystal water loss with endothermic heat effect [4]. Mansouri et al. dried the sample prepared at 30 °C overnight and found that the release of the supposed water starts above 30 °C. The mass loss is accompanied by an endotherm thermal effect until 130 °C, where an exothermic reaction occurs, which is probably a redox reaction between the ammonia ligands and permanganate ion. There was no direct evidence given that the eliminated component is only water around 100 °C. We tested the removal of water by heating with benzene as a heat-convection medium when compound **2** did not release its water as benzene-water azeotrope at 80 °C. In the solid state, heating at 100 °C resulted in water loss only on prolonged (4 h) heating (Figure S2). Compound **1**, prepared with heating of compound **2**, has a polycrystalline nature. However, compound **2** could be crystallized out as plate-like single crystals from the mother liquor of its synthesis with the use of sodium permanganate. The powder X-ray diffractogram of compound **2** (calculated from the SXRD measurements

data) fits the experimentally found powder pattern obtained for the sample of compound **2** (Figure S5).

### 2.2. Structural Features of Compound **2**

The polycrystalline nature of compound **1** did not allow us to determine its structure. However, compound **2** could be crystallized as purple plate-like orthorhombic single crystals from the mother liquor of its synthesis reaction. The main crystallographic parameters of compounds **1**, **2** (Tables S1 and 2), and **1-ClO₄** are given in Table 2. The main measured crystallographic data of compound **2** and **1-ClO₄**, including atomic parameters, bond lengths, and angles, are given in Tables 3, 4 and S2–S6.

**Table 2.** The main crystallographic parameters of compounds **1**, **2**, and compound **1**-ClO₄.

| Compound | Compound 1 | Compound 2 | Compound 1-ClO₄ [43] |
|---|---|---|---|
| **Temperature, K** | 296 | 101.1(6) | 296 |
| **Crystal system** | Triclinic | Orthorhombic | Orthorhombic |
| **Space group** | *P-1* | *Pbca* | *Pnma* |
| **Lattice constants, Å** | *a* = 7.09040(5) | *a* = 11.10180(10) | *a* = 17.8961(5) |
| | *b* = 8.77177(5) | *b* = 10.57700(10) | *b* = 8.0768(2) |
| | *c* = 9.76420(5) | *c* = 17.06750(10) | *c* = 6.8871(2) |
| **Volume, Å³** | 588.66 | 2004.13(3) | 995.48(5) |
| **$D_c$, mg/m³** | 3.106 | 1.139 | 1.147 |
| **Z** | Z = 1 | 8 | 4 |

Compound **1** is triclinic (PXRD), **1**-ClO₄ is orthorhombic (SXRD), and compounds **2** and **1**-ClO₄ are orthorhombic, but they are not isomorphous with each other. Compounds **3**, **4**, and **5** are monoclinic. Only two hydrated [carbonatotetraamminecobalt(III)] oxyanion salts, namely the nitrate (compound **4**) and sulfate (compound **5**), are known structures, and both compounds contain two different kinds of cations (conglomerate-type crystals). Compounds **1** and **2** have different unit cell parameters (Table 2). The change in the cell parameters is due to the crystal water content of compound **2**. However, after the loss of crystal water, the cell of compound **2** rearranges and transforms into a lower symmetry triclinic crystal system (Table 2).

The asymmetric unit of compound **2** contained one $[Co(NH_3)_4CO_3]^+$ cation, one $MnO_4^-$ anion, and one water molecule (Figure 1). Thus, the hydrated permanganate complex (compound **2**) contains only one kind of cation in its unit cell.

The ligands are arranged around the $Co^{3+}$ ion in a distorted octahedral geometry. The distortion of the ideal octahedral geometry is caused by the presence of the bidentate $CO_3^{2-}$ ligand, as shown by the O1-Co1-O2 angle being 68.62° instead of the ideal 90° (Table S2). This angle is similar (68.41°) in the symmetric anhydrous perchlorate complex containing carbonate ion (compound **1-ClO₄**), however, in the hydrated permanganate salt (compound **2**), similarly to other hydrated salts (compound **4** [45,46] and compound **5** [47]), the carbonate ion is bound asymmetrically to cobalt(III) (C-O1 = 1.318, C-O2 = 1.310) and a weak trans-effect can also be observed, namely, the trans-ammonia molecules oriented towards the carbonate oxygens are bound a bit stronger than the axial ammonia ligands. It might be attributed to the effects of water on the nature of non-covalent interactions, as the hydrogen bond systems are built up by the ammonia ligands and oxyanions in compound **2**.

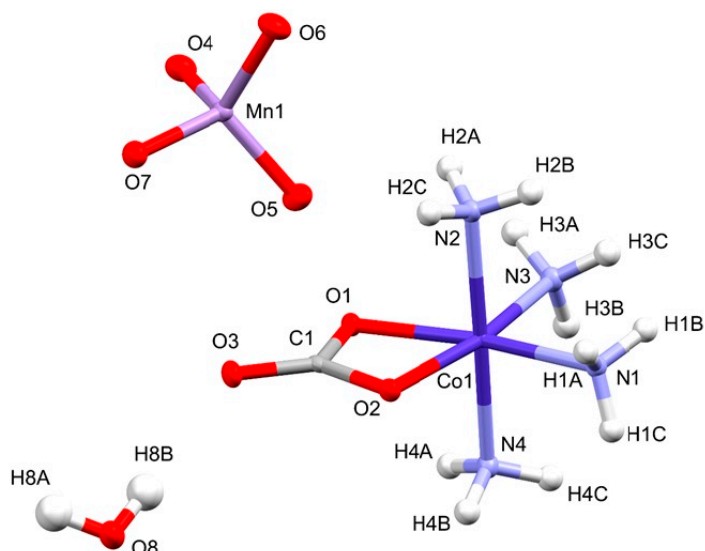

**Figure 1.** The asymmetric unit of compound **2**.

The structure of compound **2** is stabilized by a well-developed intermolecular hydrogen bond system, whereas intramolecular hydrogen bonds cannot be found. Each water molecule forms three hydrogen bonds, thereby connecting three complex cations through water-mediated hydrogen bonds (Figure 2a). Each $MnO_4^-$ anion connects six $[Co(NH_3)_4CO_3]^+$ complex cations by forming a total of seven hydrogen bonds (Figure 2b). The $CO_3^{2-}$ ligand forms a total of nine hydrogen bonds, two with two water molecules and four and three with nitrogen atoms of two neighboring $[Co(NH_3)_4CO_3]^+$ cations (Figure 2c). The four $NH_3$ ligands each form 3–5 hydrogen bonds, but the ligands differ from each other in terms of hydrogen bonding partners (Figure 2d).

The four $NH_3$ ligands form a total of 15 hydrogen bonds with six $MnO_4^-$ anions, two $[Co(NH_3)_4CO_3]^+$ complex cations, and one water molecule. The nitrogen N1 atom forms 3 hydrogen bonds with a water molecule, a $CO_3^{2-}$ ligand of a neighboring cation, and an $MnO_4^-$ anion. N2 forms five hydrogen bonds with $CO_3^{2-}$ ligands of two neighboring cations and two $MnO_4^-$ anions. N3 forms a total of four hydrogen bonds with $CO_3^{2-}$ ligands of two neighboring cations and a $MnO_4^-$ anion. N4 forms three hydrogen bonds with three different $MnO_4^-$ anions. The bond lengths and angles regarding the hydrogen bond system in the structure of compounds **1-ClO₄** and **2** are given in Tables 3 and 4, respectively.

**Table 3.** Hydrogen bond analysis of $[Co(NH_3)_4CO_3]ClO_4$ [43] (distances in Å, angles in °).

| Nr | Donor—H···Acceptor | Symm. op. | D—H | H···A | D···A | D-H···A |
|---|---|---|---|---|---|---|
| 1 | N1--H1A···O5 | x, y, z | 0.89 | 2.52 | 3.371(6) | 159 |
| 2 | N1--H1B···O2 | x, y, −1 + z | 0.89 | 2.18 | 3.017(3) | 156 |
| 3 | N1--H1C···O3 | ½ − x, −y, 1/2 + z | 0.89 | 2.58 | 3.063(3) | 115 |
| 4 | N1--H1C···O2 | −x, −1/2 + y, 2 − z | 0.89 | 2.58 | 3.313(3) | 140 |
| 5 | N1--H1C···O1 | −x, −y, 2 − z | 0.89 | 2.59 | 3.311(3) | 139 |
| 6 | N4--H2···O5 | 1/2 − x, −y, 1/2 + z | 0.75(3) | 2.44(3) | 3.145(4) | 158(3) |
| 7 | N3--H4···O1 | −x, −y, 2 − z | 0.78(3) | 2.35(2) | 3.048(2) | 151(3) |
| 8 | N3--H5···O2 | x, y, −1 + z | 0.82(4) | 2.24(4) | 3.020(4) | 158(3) |

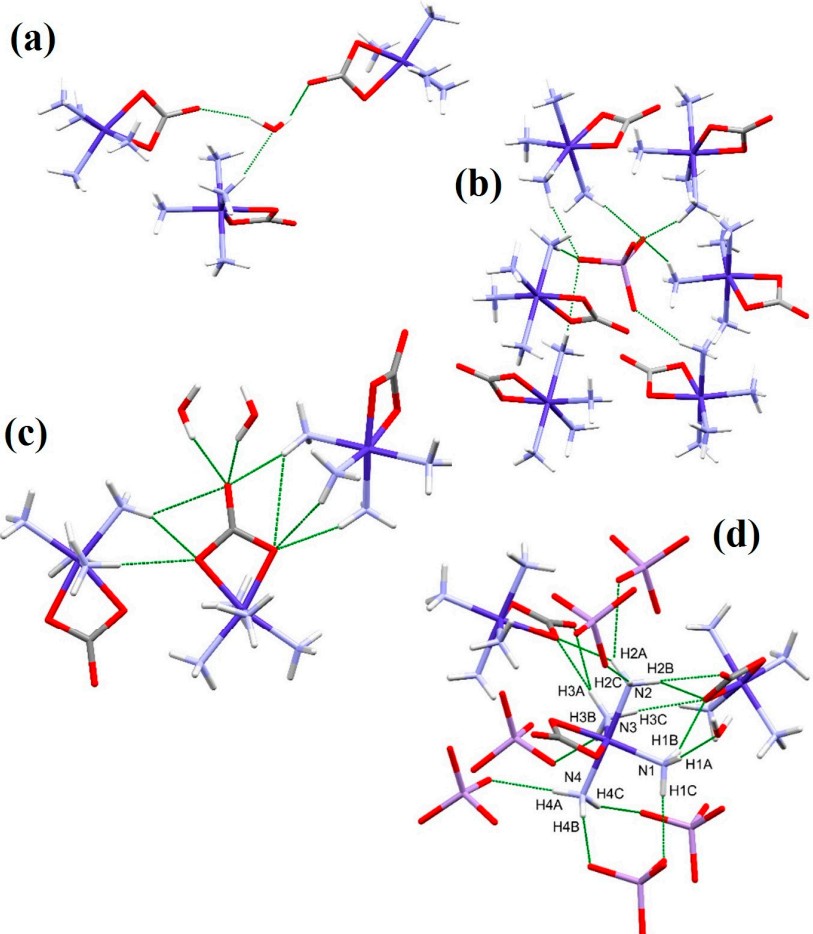

**Figure 2.** Intermolecular hydrogen bonds in the structure (in green). Hydrogen bonds formed by the water molecule (**a**), the $MnO_4^-$ anions (**b**), the $[Co(NH_3)_4CO_3]^+$ cations (**c**), and the four $NH_3$ ligands (**d**).

To gain a deeper understanding of the intermolecular interactions within the crystal structure, we performed a Hirshfeld surface analysis. The Hirshfeld surface divides the volume of the crystal into two regions in which the sum of the electron densities contributed from a selected molecule (or molecules) is larger or smaller than the contribution from all the other atoms in the crystal, respectively. The distances of the nearest internal atom to the surface ($d_i$), the nearest external atom to the surface ($d_e$), and the sum of the two values normalized by the corresponding van der Waals radii ($d_{norm}$) can be used to analyze interactions within the crystal structures (Figure 3).

Hydrogen bond interactions can be identified both on the Hirshfeld surface colored by $d_{norm}$ (Figure 3a,c,e,g,i,k) shown by red spots on the surface) and the Fingerprint plots (Figure 3b,d,f,h,j,l) as two distinct spikes for the O···H-N/O···H-O and N-H···O interactions. The percentage of N-H···O interactions towards the carbonate ligand of other complex cations or the anions on the Hirshfeld surface is 53.6% for **2** and 59.2% for compound **1-ClO₄**.

The interactions of the carbonate ligand cover 22.0% of the surface for both **2** and **1-ClO₄**. The surfaces also show that the interactions of the cation are asymmetric, as chemically identical $NH_3$ or CO groups form interactions that differ in both strength and the number and identity of their interacting partners. In general, medium-strong hydrogen bonds can be observed in both compounds, which are intermolecular ones between the ligand N-Hs and anion oxygens. These hydrogen bonds are a bit stronger in compound **2** than in compound **1**-ClO₄. The crystal water makes a stronger hydrogen bond interaction with the carbonate than the ammonia ligands do with the permanganate anion. The largest difference between the maps can be observed in Figure 3f, where the presence of water

changes the relations of O···H-O hydrogen bonds in the complex picture of O···H-N/O···H-O interactions because the O···H-O type interactions are missing in the map of compound **1-ClO$_4$**. The packing of the ions and water molecules in the crystal can be seen in Figure S6. The [Co(NH$_3$)$_4$CO$_3$]$^+$ cations and MnO$_4$$^-$ anions are arranged in alternating columns along the *b* unit cell axis (Figure S6b). The water molecules are placed between the cations and anions, whereas in the crystal of **1**-ClO$_4$, the anions and cations form a layered structure in the *cb* plane.

The hydrogen bonds are, in general, shorter and stronger in the permanganate complex (compound **2**) than in the perchlorate complex (compound **1**-ClO$_4$), resulting in a more compact structure. The Kitaigorodskii packing indexes are 69.3 and 80% for compounds **1**-ClO$_4$ and **2**, respectively. The **1**-ClO$_4$ complex contains 2.1% void volume in the crystal. The place of connection of water molecules to the carbonate ligand relative to the Hirshfeld surface can be seen in Figure 4 in comparison to the **1**-ClO$_4$ anhydrate.

Five different Co···Co distances were found between neighboring [Co(NH$_3$)$_4$CO$_3$]$^+$ cations (5.034 Å, 5.806 Å, 7.227 Å, 7.703 Å, 8.834 Å, Figure 5). For comparison in **1**-ClO$_4$, the shortest Co···Co distance is 5.119 Å.

**Table 4.** Hydrogen bond analysis of [Co(NH$_3$)$_4$CO$_3$]MnO$_4$·H$_2$O (distances in Å, angles in °).

| Nr | Donor—H···Acceptor | Symm. op. | D—H | H···A | D···A | D-H···A |
|----|--------------------|-----------|-----|-------|-------|---------|
| 1 | N1--H1A···O8 | $3/2 - x, -1/2 + y, z$ | 0.91 | 2.3 | 2.927(2) | 168 |
| 2 | N1--H1B···O4 | $3/2 - x, 1 - y, 1/2 + z$ | 0.91 | 2.53 | 2.967(2) | 110 |
| 3 | N1--H1B···O2 | $3/2 - x, -1/2 + y, z$ | 0.91 | 2.21 | 3.075(2) | 158 |
| 4 | N1--H1C···O7 | $x, 3/2 - y, 1/2 + z$ | 0.91 | 2.13 | 2.993(2) | 158 |
| 5 | N2--H2A···O1 | $1 - x, 1 - y, 1 - z$ | 0.91 | 2.38 | 3.207(2) | 151 |
| 6 | N2--H2A···O7 | $3/2 - x, -1/2 + y, z$ | 0.91 | 2.59 | 3.229(2) | 127 |
| 7 | N2--H2B···O3 | $3/2 - x, -1/2 + y, z$ | 0.91 | 2.08 | 2.986(2) | 171 |
| 8 | N2--H2C···O5 | $x, y, z$ | 0.91 | 2.10 | 2.960(2) | 157 |
| 9 | N3--H3A···O1 | $1 - x, 1 - y, 1 - z$ | 0.91 | 2.14 | 3.004(2) | 158 |
| 10 | N3--H3A···O3 | $1 - x, 1 - y, 1 - z$ | 0.91 | 2.59 | 3.377(2) | 145 |
| 11 | N3--H3B···O4 | $1 - x, 1 - y, 1 - z$ | 0.91 | 2.20 | 3.051(2) | 156 |
| 12 | N3--H3C···O2 | $3/2 - x, -1/2 + y, z$ | 0.91 | 2.24 | 3.131(2) | 165 |
| 13 | N4--H4A···O7 | $-1/2 + x, 3/2 - y, 1 - z$ | 0.91 | 2.15 | 3.056(2) | 175 |
| 14 | N4--H4B···O4 | $x, 3/2 - y, 1/2 + z$ | 0.91 | 2.17 | 3.079(2) | 172 |
| 15 | N4--H4C···O6 | $3/2 - x, 1 - y, 1/2 + z$ | 0.91 | 2.17 | 3.051(2) | 164 |
| 16 | O8--H8A···O3 | $1 - x, 2 - y, 1 - z$ | 0.87 | 1.99 | 2.836(2) | 165 |
| 17 | O8--H8B···O3 | $x, y, z$ | 0.87 | 2.04 | 2.825(2) | 150 |

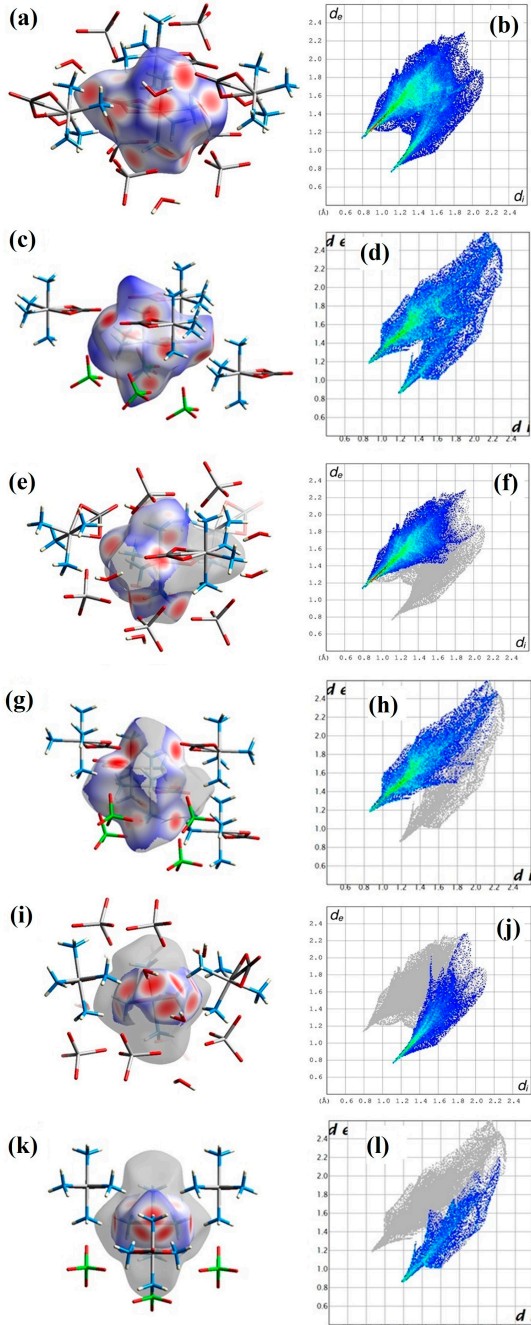

**Figure 3.** Hirshfeld surface analysis of the $[Co(NH_3)_4CO_3]^+$ cation in compound **2** and **1**-ClO$_4$. The Hirshfeld surface of the complex cation is colored by $d_{norm}$ with contacting molecules shown as sticks in (**a**) and (**c**), and the corresponding Fingerprint plots are shown in (**b**) and (**d**) for compounds **2** and **1**-ClO$_4$, respectively. N-H⋯O interactions of the cation are shown on its Hirshfeld surface (**e**) and (**g**) and in the corresponding Fingerprint plot (**f**) and (**h**) for compounds **2** and **1**-ClO$_4$, respectively. O⋯H-N and O⋯H-O interactions of the cation are shown on its Hirshfeld surface (**i**) and (**k**) and in the corresponding Fingerprint plot (**j**) and (**l**) for compounds **2** and **1**-ClO$_4$, respectively. Regions, where the sum of $d_i$ and $d_e$ is smaller (stronger interaction), equal to, or larger (weaker interaction) than the sum of the van der Waals radii of the corresponding atoms are colored red, white, and blue on the Hirshfeld surface, respectively. The Fingerprint plots are colored according to the relative area of the corresponding $d_i$–$d_e$ pair on the Hirshfeld surface from white (no contribution) through blue (small contribution) to green and red (large contribution).

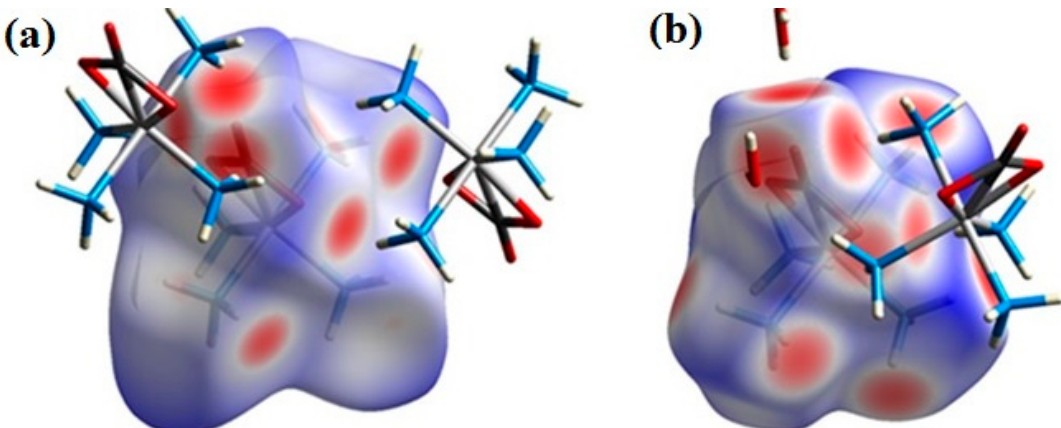

**Figure 4.** Comparison of the secondary interactions of the carbonate ligand in the case of compound **1**-ClO$_4$ (**a**) and **2** (**b**) shown on Hirshfeld surfaces, highlighting the part of the complexes where the water molecules are bound in **2**.

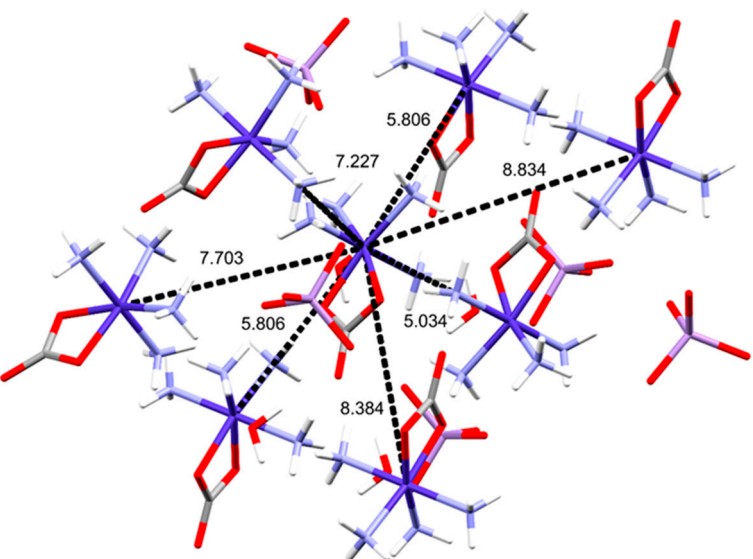

**Figure 5.** Distances between neighboring Co atoms in the structure.

### 2.3. Spectroscopic Properties of Compounds **1** and **2**

The room-temperature vibrational spectroscopic (IR, far-IR, and Raman) and low-temperature Raman results (liq. N$_2$ temperature) of compounds **1** and **2** are given in Tables 5, 6, S7 and S8. Correlation analysis for the vibrations in the unit cell was performed on ammonia, carbonate, cobalt, permanganate, and water species located in the structure of compound **2**. Each vibrational mode belonging to a cis-O$_2$CoN$_4$ skeleton, ammonia, and carbonate-ion ligands, and the outer sphere permanganate ion was assigned with the use of correlation method of vibrational analysis and the available IR and Raman data of the known [carbonatotetraamminecobalt(III)] complex salts [48,52–54] including theoretical spectroscopic considerations on the [Co(NH$_3$)$_4$CO$_3$]$^+$ ion were also considered [53,54]. The correlation analysis results showed that altogether, there are 312, 168, and 144 internal, hindered translational, and hindered rotational (librational) modes, respectively, belonging to compound **2**. Three modes (B$_{1u}$, B$_{2u}$, and B$_{3u}$) of all hindered translations are due to lattice acoustic modes, leaving a total of 165 hindered translational optical modes. There are 29 atoms in the formula unit, which means (due to Z = 8) there are 232 atoms in the unit cell. The latter being primitive, there are 696 degrees of freedom. This is in accord with the sum of internal vibrations (336), hindered translations (192), and hindered rotations (168) = 696, as it must be. Overlapped bands due to the external modes are expected a priori.

**Table 5.** The internal ($\nu_1$–$\nu_4$) and rocking ($\nu_5$) [55] vibrational modes of ammonia ligands in the IR spectra of compounds **1** and **2**.

| Mode | $[Co(NH_3)_4CO_3]^+$, $cm^{-1}$ | Measured Values in $cm^{-1}$ at Room Temperature | | | | Assignment |
|---|---|---|---|---|---|---|
| | Ref. [55] | Our Results | | | Ref. [48] | |
| | | **1** | **2** | **2-D** | **5** | |
| $\nu_1$ | - | 3242, 3168 | 3151 | 2347sh, 2333sh, 2303 | 3289, 3192 | $\nu_s$(N-H) |
| $\nu_2$ | 1300 | 1298 | 1346 | 997 | 1300 | $\delta_s$(H-N-H) |
| $\nu_3$ | - | 3288 | 3281 | 2462, 2450 | 3425 | $\nu_{as}$(N-H) |
| $\nu_4$ | 1638 * | 1579 * | 1621 * | 1187, 11154 | 1645 * | $\delta_{as}$(H-N-H) |
| $\nu_5$ | 810 | ~800 | ~800 | 664, 651 | 828 | $\rho$(NH$_3$) |

* overlapped bands; calculated from the $\delta_s$(D-N-D) value with the $\nu$(N-H)/$\nu$(N-D) = 1.35 ratio.

## 2.4. Vibrational Modes of the Ammonia and Carbonate Ligands in Compounds **1** and **2**

There is only one type of complex cation located in the unit cell of compound **2** (different from the analogous sulfate and nitrate compounds). Thus, four crystallographically different ammonia ligands and one type of carbonate ion can be distinguished. Each ammonia ligand under $D_{2h}$ symmetry has four internal modes ($\nu_1$, symmetric stretching, $\nu_2$, symmetric bending, $\nu_3$, antisymmetric stretching, and $\nu_4$, antisymmetric bending) and six external (three hindered rotational and three hindered translational) vibrational modes. All modes are active in the IR and Raman spectra. The number of internal vibration modes for ammonia ligands is 48 and must be multiplied by four due to the four crystallographic types of ammonia ligands. For the same reasons, the number of external modes for one type of ammonia ($24T + 24R$) is also multiplied by four (Figure S7).

The NH$_3$ and H$_2$O vibrational modes in the Raman spectra of compounds **1** and **2** measured with 785 nm laser excitation are not intensive enough to evaluate completely (Figures S8 and S9), and the measuring range (2000–200 cm$^{-1}$) does not contain the stretching mode region of N-H and O-H bonds. On the other hand, the excitation with a 532 nm laser, which was used between 4000 and 100 cm$^{-1}$ and resulted in a better signal/noise ratio for N-H and O-H vibrational modes for similar compounds [39,48], led to the complete decomposition of compounds **1** and **2** even with cooling to −150 °C and decreasing the intensity of the laser energy to the minimum attainable value. This strongly suggests that the energy of the 532 nm laser is large enough to initiate the redox reaction between the ammonia ligands and permanganate anion. However, in the IR spectrum recorded at room temperature (Figure S10, Table 5), all vibrational modes of ammonia ligands, including the rocking ($\nu_5$) modes of NH$_3$ molecules [7,8], were identified.

Mansouri studied the IR spectra of compound **1** and assigned the $\nu$(N-H) modes at 3301 cm$^{-1}$ and declared that the band observed at 3400 cm$^{-1}$ belonged to the adsorbed water. According to our results (Figure S10b), the stretching modes of OH bands belonging to water (compound **2**) and N-H bands (compounds **1** and **2**) belonging to ammonia can easily be separated. To distinguish some overlapping bands as $\nu$(C=O) and $\delta$(H-N-H) and assign each N-H mode, the anhydrous compound **1** was digested in D$_2$O and prepared the partially deuterated (D/H ratio ca. 4:1) compound **2** called to be **2-D**. The intensity of water O-H/O-D and N-H/N-D band ratios together with the important shifts of O-D and N-D band positions (the band positions belonging to groups which are not connected directly to hydrogen(deuterium) were either not or only slightly shifted) give a sensitive tool to assign the modes and bands belonging to hydrogen (deuterium) containing groups. Taking into consideration the experimental results (Table 5 and Figure S10c), the O-D and N-D bands could easily be assigned. The designation $\nu$(C=O) comes out due to the symmetry descent

of the carbonate anions (they are bidentate coordinated to two Co$^{III}$ cations), so the bond order of the uncoordinated C=O group is close to 2.

The weak band at 1658 cm$^{-1}$ of compounds **1** and **2** does not alter on deuteration. Thus, this band cannot belong to hydrogen-containing groups (H$_2$O or NH$_3$). The complex band centered at ~1620 cm$^{-1}$ contains at least two bands, $\nu$(C=O) and $\delta_{as}$(H-N-H), for compounds **1**, and three bands, $\nu$(C=O), $\delta_{as}$(H-N-H) and $\delta_{as}$(H-O-H) for compound **2**. The position of the band component is located at 1590 cm$^{-1}$ in the IR spectrum of compound **2-D** and appears as a shoulder in the IR spectra of each of the compounds **1** and **2**. Thus, this band is unambiguously assigned as the $\nu$(C=O) component. The band centered at 1620 cm$^{-1}$ in the spectra of hydrated compound **2** coincides with a shoulder located on the left side of the complex band in the IR spectra of compound **1** (Figure S11), whereas there are other coinciding shoulders on the right side of the complex bands in the spectra of compounds **1** and **2**. Thus, the ammonia bands are located at the right side of the $\nu$(C=O) carbonyl band, whereas the band at ~1620 cm$^{-1}$ found in the spectrum of compound **2** is mainly due to $\delta_{as}$(HOH). Since a shoulder appeared in the IR spectrum of compound **1** as well on the left side of the carbonyl band, a band component ($\delta_{as}$(HNH)) is present in this complex band in the IR spectra of compound **1**. Thus, the scissoring mode of water is a dominant component on the left side shoulder of the complex band in the spectrum of compound **2**, together with a small contribution of the $\delta_{as}$(HNH) component. The appearance of two new peaks in the spectrum of **2-D** at 1187 cm$^{-1}$ and 1152 cm$^{-1}$ confirms the presence of non-equivalently deuterated ammonia ligands. The theoretical positions of N-H bands in the spectrum of compounds **1** and **2**, taking into consideration the $\nu$(N-H)/$\nu$(N-D) position ratio of 1.35 are expected to be located around 1602 and 1558 cm$^{-1}$, whose values are close to the experimentally found values for compounds **1** and **2** (Figure S11). The well-defined band in the IR spectrum of compound **2-D** at 997 cm$^{-1}$ belongs to the $\delta_s$(D-N-D). Accordingly, the intensity of the band found at ~1300 cm$^{-1}$ in the IR spectrum of compound **2-D** compared with the intensity of the band in the IR spectrum of compounds **1** or **2** decreased with decreasing amount of the N-H groups in the sample. There are complex bands in the spectra of compounds **1** and **2**, which consist of at least two components. The second component at 1278 cm$^{-1}$ shows no shift on deuteration; thus, that band does not belong to ammonia but carbonate ion, whereas the band at 1298 cm$^{-1}$ almost completely disappeared, with the simultaneous appearance of the $\delta_s$(D-N-D) at 967 cm$^{-1}$. The $\nu$(N-H)/$\nu$(N-D) ratio was found to be 1.34, which agrees well with the expected value (1.35). The comparison of these bands in the IR spectra of compounds **1** and **2** shows that some positions of N-H bands are not the same in compounds **1** and **2**. In general, the N-H bands are shifted to the lower wavenumbers, probably due to the presence of hydrogen bonds between the ammonia ligands and water molecules. The rocking mode is not as sensitive to deuteration as the deformation and stretching modes because of the different kinds of motions during the vibration. Accordingly, the wide band system was assigned to the rocking mode of ammonia ligands, whose intensity substantially decreased, and a multicomponent wide band system appeared with peaks at 664 and 651 cm$^{-1}$ (Figure S11).

The isolated carbonate ion is planar ($D_{3h}$), accordingly, four internal normal modes of vibration: symmetric stretching, $\nu_1(A_1')$, $\nu_s$(CO$_3$); symmetric bending, $\nu_2(A_2')$, $\pi$(CO$_3$); antisymmetric stretching ($\nu_3(E')$, $\nu_{as}$(CO$_3$), and antisymmetric bending $\nu_4(E')$, $\delta_{as}$(OCO) are expected to occur. The $\nu_1$, $\nu_3$, and $\nu_4$ are IR and Raman active modes, but due to the lifting of the degeneracy upon coordination, all modes are expected to be IR and Raman active in compounds **1** and **2**. The correlation method vibrational analysis results for the $D_{2h}$ carbonate ion in compound **2** can be seen in Figure S12.

There are 24 IR and 24 Raman active internal modes of vibration, four $\nu_1$ and four $\nu_2$ (symmetric stretching and bending, respectively) modes, and eight $\nu_3$ and eight $\nu_4$ (antisymmetric stretching and bending, respectively) in both IR ($A_g$, $B_{1g}$, $B_{2g}$, $B_{3g}$) and Raman ($A_u$, $B_{1u}$, $B_{2u}$, and $B_{3u}$). It means 48 internal modes of vibration, and there are altogether 48 external vibrational modes (24$T$ + 24$R$) as well (Figure S12).

The bidentate coordination of carbonate ion gives (as mentioned earlier) two kinds of carbon-oxygen bonds, namely, the C=O$^\S$ non-coordinated bonds, and two C-O bonds with coordinated oxygen atoms. The IR data of compounds **1** and **2** together with the data of **1**-ClO$_4$ can be seen in Table 6.

**Table 6.** IR wavenumbers of carbonate ion modes in compounds **1** and **2**.

| Band | [Co(NH$_3$)$_4$CO$_3$]$^+$, Calculated, cm$^{-1}$ [54,55] | Measured, cm$^{-1}$ | | |
|:---:|:---:|:---:|:---:|:---:|
| | | **1** | **2** | **1-ClO$_4$** |
| $\nu$(C=O)$^\S$ | 1577 | 1590 * | 1590 * | 1602 |
| $\nu_s$(C-O) | 1052 | 1026 | 1035 | - |
| $\delta$(O-C-O), in plane | 771 | 757 | 754 | 762 |
| $\nu_{as}$(C-O) | 1274 | 1278 * | 1278 * | 1284 |
| $\delta$(O-C=O$^\S$), in plane | 671 | 675 | 672 | 672 |
| $\pi$, out of plane | 859 | 824 | 828 | 836 |

* Component of a complex band system.

The in-plane $\delta$(O-C-O) and $\delta$(O-C=O$^\S$) deformation bands contain some contribution of $\nu$(Co-O) [55]. Furthermore, the mutual contribution of $\nu_{as}$(C-O) and $\delta$(O-C=O$^\S$) could also be detected by Fujita in the case of coordinated carbonate complexes [55]. The deuteration had no important influence on the positions of the carbonate ion bands in the IR spectra of compounds **2** and **2-D**, as was found in the IR spectra of **1**-ClO$_4$ and its deuterated form [55]. The difference between the symmetric and antisymmetric stretching C-O modes in compounds **1** and **2** ($\Delta\nu$ = 252 cm$^{-1}$ and 243 cm$^{-1}$, respectively) shows that the carbonate ions are coordinated in both compounds similarly, namely, as a chelating bidentate ligand [54,56]. The Raman bands of carbonate ions in the Raman spectra of compounds **1** and **2** are very weak. Thus, further information about the nature of coordinated carbonate ions could not be gained from the Raman spectra of compounds **1** and **2**. However, the Mn-O modes appeared in the Raman spectra with considerable intensities. Thus, the Raman spectroscopic properties of permanganate ion in compounds **1** and **2** were evaluated in detail (see below).

### 2.5. Vibrational Modes of the cis-CoN$_4$O$_2$ Core in Compounds **1** and **2**

The strongly distorted octahedral cation in compounds **1** and **2** contains cis-CoN$_4$O$_2$ units. The total number of the external modes of Co$^{III}$ central ions in compound **2** is 24 (12 IR and 12 Raman), and there are hindered translational modes ($Z$ = 8). Three kinds of stretching and bending modes can be used for a cis-O$_2$CoN$_4$ cation, namely, NCoN, NCoO, and OCoO, which are expected to be considerably coupled [53]. The measured far-IR (Figure S13) and Raman data and the calculated parameters ($f_{CoN}$ = 1.6 and $f_{CoO}$ = 1.25) for the isolated [Co(NH$_3$)$_4$CO$_3$]$^+$ ion [53] are given in Table S7. Based on the common assumption that M-N stretching frequencies are at higher wavenumber than the M-O stretching, logically, the $\nu_s$(NCoN) > $\nu_s$(NCoO) > $\nu_s$(OCoO) order and $\nu_{as}$ > $\nu_s$ relationship can be taken into consideration [42]. The cis-CoO$_2$ moiety in carbonate-chelated tetraamminecobalt(III) complexes have higher experimental antisymmetric than symmetric Co-O mode wavenumbers; thus, the assignments of CoO$_2$ stretching modes are given in Table S8 accordingly [54,56]. It is not possible to unambiguously distinguish the Co-N bands belonging to CoN$_2$ from those of CoNO moieties due to the symmetry lowering of octahedral geometry, the asymmetrically coordinated carbonate group, the differences in Co-N and Co-O bond distances, and consequently due to the mixed character of these bands (Figure S14) [53].

There is a wide band in the IR and a three-component band system in the Raman spectra of compounds **1** and **2** (Figures S8–S10), which contains the $\nu_6$ (antisymmetric Co-O stretching) component of cis-O$_2$CoN$_4$ core and two, (probably a doublet) antisymmetric

Mn-O deformation of the permanganate ions. Since the $\nu_4$ ($\delta_{as}$) bands of permanganate ions are always in the lower-frequency IR region, and the two lower band intensities in the Raman spectra are comparable, whereas the first component has a higher intensity than that of the other two bands, it is possible that the highest wavenumber component contains the $\nu_{as}$(Co-O) mode mixed with the $\delta_{as}$(Mn-O) mode whereas the two other two belong only to the $\delta_{as}$(Mn-O) mode (Figure S15).

Comparison of the Raman spectra of compounds **1** and **2** in the far-IR region (Figure S15) shows that the bands belonging to the cis-$CoO_2N_4$ core are very similar in both compounds, although some symmetry changes (splitting of Co-N stretching bands) could be observed. This shows that the crystal structure and the polarization of Co-N bonds changed during the dewatering of compound **2** into compound **1**. The estimated metal-ammonia bond strength parameter ($\varepsilon$) calculated from the positions of $\delta_s$(H-N-H) bands shows 0.59 and 0.78 values for compounds **1** and **2**, respectively [57,58]. This confirms that the average strength of the ammonia coordination (consequently, the Co-N average bond length) is not equal in the two compounds. The average bond strength is higher (the average bond length is shorter) in compound **2** than in compound **1**.

### 2.6. Vibrational Modes of the Non-Coordinated Species (Permanganate Ion and Crystal Water) in Compounds **1** and **2**

An isolated permanganate ion under $T_d$ has four normal modes (two symmetric and two antisymmetric modes), two stretching, and two bending modes. All four modes of the ideally symmetric anion are Raman active, whereas only the antisymmetric ones (stretching and bending) are IR active. The antisymmetric mode is trip, whereas the symmetric bending mode is doubly degenerate [5–7]. The permanganate ion in compound **2** is isolated but distorted ($D_{2h}$ factor group symmetry) due to its crystallographic position and the hydrogen bonds of its oxygens. There are 72 internal (36IR and 36Raman) and 48 external (12 Translational + 12 Rotational both in IR and Raman) modes (Figure S16).

The IR forbidden bands become IR active due to distortion of the tetrahedral permanganate anion, but the intensities of these bands are weak in the IR spectra of compounds **1** and **2** (Table S8 and Figures S10 and S11). The $\delta_s$(Mn-O) could not be detected in the IR spectrum of compound **1** due to its low intensity. However, it appears as a weak shoulder in the IR spectrum of compound **2**. The Raman intensity of the symmetric stretching mode ($\nu_s$), however, is the most intense band of the Raman spectra, which unambiguously assigns this mode at 836 cm$^{-1}$ for both compounds.

The intense $\nu_s$ band in the Raman spectra of compounds **1** and **2** at ~836 cm$^{-1}$ appears as a singlet (Figures S8 and S9, respectively). The $\nu_{as}$ bands in the Raman spectra of compounds **1** and **2** split into 3 and 4 components at 298 K and 123 K, respectively. Two and three components as symmetric and antisymmetric deformation mode bands could be detected in the Raman spectra of both compounds at both temperatures (Table S8). The highest wavenumber component of the $\delta_{as}$ mode is probably mixed with $\nu_{as}$(Co-O) mode (see above). The IR spectra of these compounds showed wide singlet-like $\nu_{as}$ bands without any splitting. However, the multi-component nature of this complex band is shown by the number of the possible combinations ($\nu_1 + \nu_{3a,b,...}$) and overtone ($2 \times \nu_{3a,b,...}$) bands observed between 1700 and 1800 cm$^{-1}$ [57,59].

There are numerous (1730, 1747, 1765, ~1800 (wide) 1825 and 1839 cm$^{-1}$) and (1750, 1765, 1788, 1799, 1808, 1825, 1840 cm$^{-1}$) overtone and combination (i.e., two-photon) bands for compounds **1** and **2**, respectively, which confirms the existence of more than two $\nu_{3a,b,...}$ components in the complex $\nu_3$ band. The complex unresolved band around 800 cm$^{-1}$ found in the IR spectrum of compound **1** is split into three components in the spectra of compound **2** (Figure S11).

There are three internal vibrational modes of water under $C_{2v}$ symmetry, namely $\nu_1$ (symmetric stretching mode, $\nu_s$), $\nu_2$ (symmetric bending mode, $\delta_s$), and $\nu_3$ (antisymmetric stretching mode, $\nu_{as}$). The total number of internal vibrations is 12IR and 12Raman, and there are 24 modes altogether. According to 3 hindered rotations and 3 hindered

translations, there are 24 translational and 24 rotational modes of external vibrations. The hindered rotations of the water molecules are designated as rocking = $\rho$ = $R_x$ = in-plane libration; wagging = $\omega$ = $R_y$ = out-of-plane libration; twisting = $\tau$ = $R_z$ = out-of-plane libration (Figure 6).

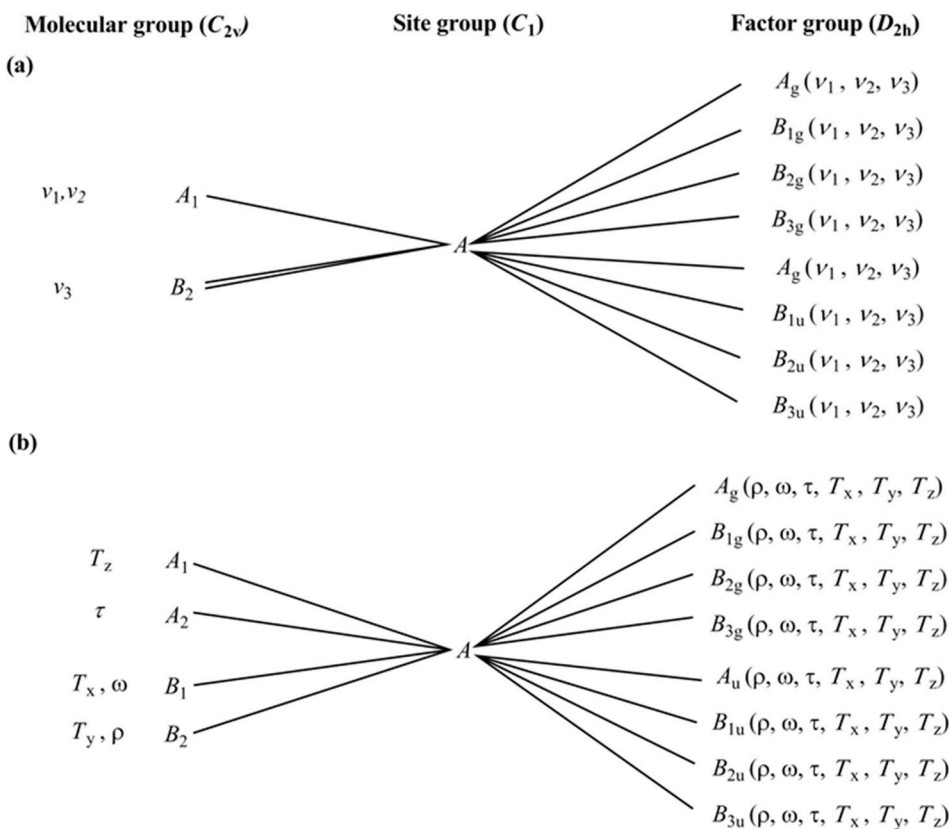

**Figure 6.** Correlation analysis diagram (**a**) internal and (**b**) external modes for crystallization water in compound **2**.

The stretching O-H modes (antisymmetric and symmetric) are located as a pair of bands with peaks at ~3512 and 3455 cm$^{-1}$, respectively. The deformation mode of water (scissoring mode) is located as a component of a mixed band containing the $\delta_{as}$(HNH) mode as well see above). The deuteration experiments resulted in D$_2$O containing hydrated compound (**2-D**), and its antisymmetric and symmetric O-D stretching modes appear at 2629 cm$^{-1}$ and 2535 cm$^{-1}$, with the ratios of $\nu$(O-H)/$\nu$(O-D) 1.34 and 1.36, respectively. A new band belonging to the O-D antisymmetric deformation mode appeared at 1191 cm$^{-1}$. It means that the $\delta_{as}$(H-O) band may be expected to appear around 1620–1600 cm$^{-1}$, exactly at that wavenumber region where we assigned it as part of the overlapped band system component found in the IR spectrum of compound **2** (Figures S10 and S11).

*2.7. UV-Vis Spectroscopy*

The solid phase diffuse reflection UV spectra of compounds **1** and **2** consist of transitions belonging to the carbonatotetramminecobalt(III) cation and permanganate ion. The diamagnetic (low-spin) Co$^{III}$ ion ground state electron configuration is $t_{2g}^6$ ($^1A_{1g}$). The electron excitation from the $t_{2g}$ orbital to the $e_g$ orbital results in the $t_{2g}^5 e_g^1$ configuration ($^3T_{1g}$ + $^1T_{1g}$ + $^1T_{2g}$ + $^3T_{2g}$ states). Both triplet states lie at energy levels lower than that of the singlet, and Sastri found a significant mixing of $t_{2g}^5 e_g^1$ and $t_{2g}^4 e_g^2$ levels and $\sigma^*$ character of the $e_g$ levels in carbonatotetraamminecobalt(III) complexes [55].

Among the four possible transitions for the [Co(NH$_3$)$_4$CO$_3$] cation [56,60], the bands at 391/390 nm and 670/620 nm were assigned together with the $\pi$-$e_g$ LMCT transition at

250/2252 nm for compounds **1** and **2**, respectively. The most intense band of this cation ($^1T_{1g} \leftarrow {}^1A_{1g}$) coincides with the most intense band of permanganate ion (($^1A_1 - {}^1T_2$)($t_1 - 2e$)) resulting in a complicated band system with multiple components peaked at 560/550 and 521/504 for compounds **1** and **2**, respectively (Figure S17) [61,62]. The presence of the L($\pi$)$\rightarrow$Co($e_g$) MLCT band was confirmed by the linear relationship between the singlet transitions wavenumbers and CT band positions given for carbonatocobalt(III) complexes. This shows that the covalency of the carbonate ions in compounds **1** and **2** are close to each other and >50% (the covalency of the carbonate group derived from the $\nu$(C=O) IR band position of these complexes is ~44% for both complexes) [62,63]. There are two spin-allowed transitions for the complex cation in compounds **1** and **2**, but the expected $^3T_{1g} \leftarrow {}^1A_{1g}$ transition is out of our measurement range (>800 nm). The differences between the energy values of the $^3T_{2g} \leftarrow {}^1A_{1g}$ transitions for compounds **1** and **2** (670 and 620 cm$^{-1}$) show different cation-anion (hydrate water) interactions in the two compounds (Figure S17).

*2.8. Thermal Decomposition of Compound* **2**

The thermal decomposition of a multiphase mixture containing mainly compound **1** was studied by Mansouri et al. [4] and detected exothermic decomposition steps, which strongly suggests that a solid phase quasi-intramolecular redox reaction takes place as was observed for some other permanganate salts of amminecobalt(III) complexes [41,42]. Since compound **2** was thermally dehydrated into compound **1**, the detailed TG-MS and DSC studies were conducted only on compound **2** (compound **1** is its decomposition intermediate) both in an inert (when only the anion-ligand redox reaction can occur) or in an oxygen-containing atmosphere (when external oxygen as co-oxidant may also take place in the redox interactions) (Figures S18–S20).

The first thermal decomposition step of compound **2** in an inert atmosphere showed 5.60% weight loss, which agreed well with the loss of one molecule of water (the theoretical weight loss for one molecule of water is 5.57%) (Figure S18). The process is endothermic; the DTG and DTA peak temperatures are 100 and 105 °C, respectively. TG-MS shows that only water ($m/z$ = 18) evolves; the intensity ratio of $m/z$ = 18, 17, 16, and 15 peaks [57] shows the lack of ammonia evolution. No ammonia oxidation products ($m/z$ = 44 (N$_2$O$^+$), $m/z$ = 30 (NO$^+$), or $m/z$ = 28 (N$_2^+$)) were detected in this decomposition step. There was no oxygen evolution ($m/z$ = 32) either (Figure 7).

Keeping compound **2** in open air, the first mass loss slightly increased due to the hygroscopicity of compound **2**. This might be attributed to the presence of large enough voids in the crystal lattice to accommodate extra water molecules in addition to the crystal water. The DSC in an inert atmosphere shows an endothermic process with 109.6 °C peak temperature and 56.36 kJ/mol heat effect (Figure S20b). Under an air atmosphere, a very similar decomposition process was observed (Figure S20a), and the shape of the TG decomposition curve is the same as that in an inert atmosphere. The small increase in the weight loss of compound **2** in an air atmosphere as compared to that in an inert atmosphere may be attributed to the presence of the adsorbed water in the sample kept in an air atmosphere, which was confirmed by the TG-MS curve (Figure S19). The heat effect of the dehydration process in air was found to be 54.95 kJ/mol (Figure S20a).

The second DSC decomposition step of compound **2** (the first decomposition step of compound **1**) consists of at least two consecutive/coinciding processes at 160 °C and 162 °C under N$_2$ or O$_2$, respectively (Figure S20). The decomposition process is exothermic, which strongly suggests that a redox reaction with the oxidation of ammonia occurs. In the absence of oxygen, the permanganate ion or Co$^{III}$ may act as an oxidant. The TG-MS in an inert atmosphere shows the presence of water ($m/z$ = 18), and since the crystallization water was released even at ~100 °C, this water can only be the oxidation product of ammonia being the only hydrogen source. The oxygen can come only from permanganate ions; thus, the reaction is definitely an interaction between the ammonia ligand and permanganate anion. It is confirmed by the IR spectra of the decomposition products of compound **2**

at 100 and 160 °C (these decomposition products are called I-100 and I-160, respectively), where the spectrum of I-160 shows the absence of any permanganate ion (Figure 8).

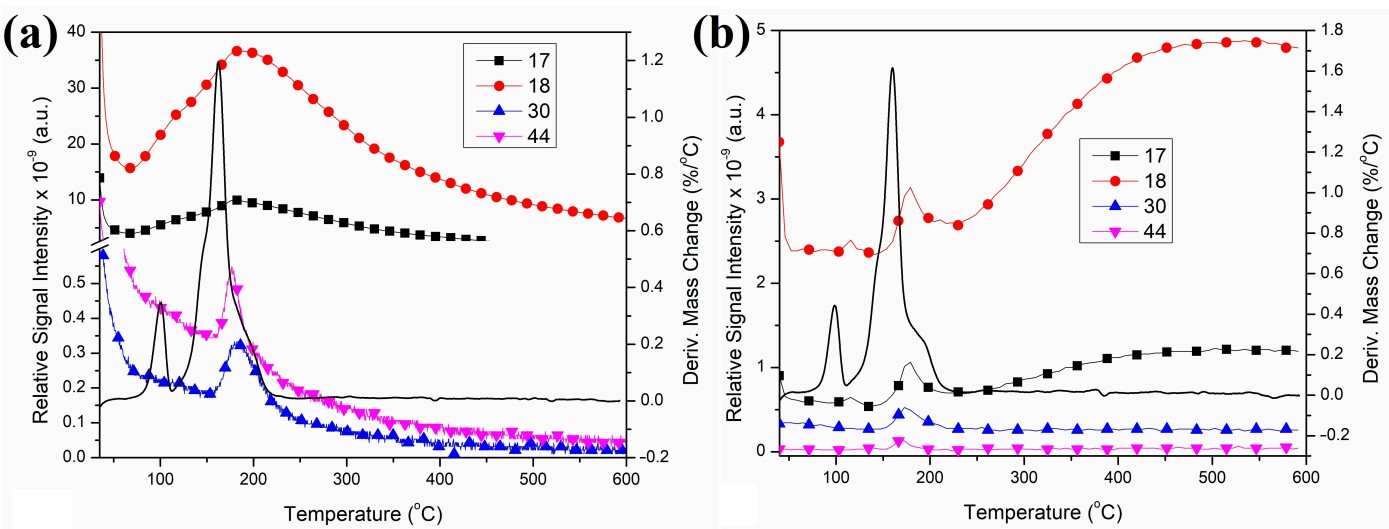

**Figure 7.** TG-MS curves of $m/z$ = 18, 17, 16, 30, and 44 ions during the thermal decomposition of compound **2** in air (**a**) and inert (**b**) atmosphere.

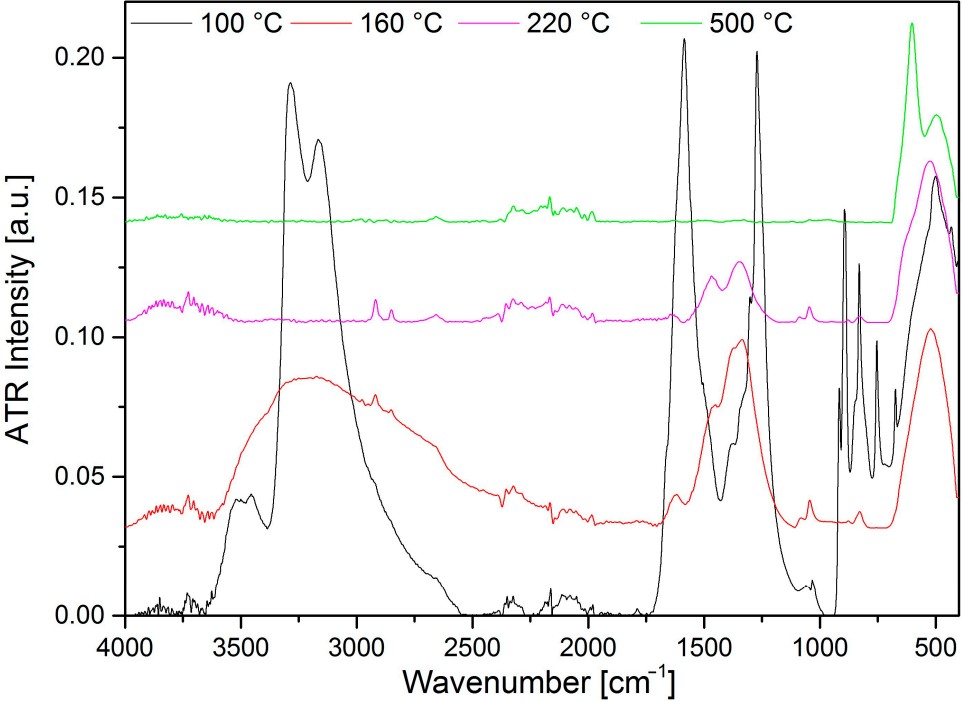

**Figure 8.** IR spectra of the thermal decomposition products of compound **2** under an inert atmosphere (I-100 (compound **1**), I-160, I-200, and I-500).

The peak intensity ratio of $NO^+/N_2O^+$ in an inert atmosphere (>1) shows that the NO is the oxidation product of ammonia, whereas, in an air atmosphere, the intensity of $m/z$ = 44 peak is higher than $m/z$ = 30 ($NO^+$). Thus, $CO_2$ formation may also play a role in the evolution of the $m/z$ = 44 peak. Since the peak at $m/z$ = 44 may belong to $CO_2^+$ and to $N_2O^+$ as well, and both species have $m/z$ = 28 and $m/z$ = 16 fragments ($N_2/CO^+$ and $O^+$, respectively), it is not possible to separate the contributions of $N_2O$ (as ammonium nitrate decomposition product) and $CO_2$ (as carbonate decomposition product). The IR spectrum of the decomposition intermediate formed at 160 °C shows

the presence of residual carbonate ion with $\nu_s$(C-O) at 1042 cm$^{-1}$, $\nu_{as}$(C-O) in the range 1406–1334 cm$^{-1}$ (overlapped with a nitrate band located at ~1380 cm$^{-1}$ [40], and $\delta$(out of plane deformation) at 826 cm$^{-1}$ (Figure 8). The position of the carbonate ion and the lack of high wavenumber C=O stretching mode bands in the spectrum of product I-160 show that the carbonate coordinated originally as a chelate became simple ionic due to the destruction of the original coordination sphere around Co$^{III}$. Thus, the signal $m/z$ = 44 mainly belongs to N$_2$O$^+$. The lack of $m/z$ = 32 discloses that $m/z$ = 16 (O$^+$ or NH$_2^+$) arises from O$_2^+$, but NH$_3^+$ ($m/z$ = 17), N$_2$O$^+$ ($m/z$ = 44) or NO$^+$ ($m/z$ = 30) may also be parent ions. The ratio of $m/z$ = 18, 17, and 16 shows that $m/z$ = 17 and 16 are not only H$_2$O$^+$ fragments, but NH$_3$ also evolves (Figure 7). The reaction heats are −170.93 and −154.02 kJ/mol in an N$_2$ and air atmosphere, respectively (Figure S20). The difference might be attributed to the fact that the oxidation reaction in an oxygen-containing atmosphere is partly covered by the elemental oxygen. The decomposition products are not the same when only the permanganate ion is the only oxidizing agent or if this role is shared between the permanganate and the excess atmospheric oxygen. If the excess (elemental) oxygen is present, the conversion of ammonia is higher, resulting in a higher reaction heat.

The decomposition intermediates in an inert atmosphere and in air decompose in endothermic processes with 171.1 and 167.2 °C peak temperatures and 148.65 and 58.96 kJ/mol heat effects, respectively. These decomposition products decompose further with 203.8 and 197.5 °C peak temperatures in exothermic processes, accompanied by −15.71 and −195.80 kJ/mol heat effects (Figure S20). For the overall process, the molar ratio of ammonia to permanganate (=4) shows that the oxidation of ammonia with the oxygen content of the permanganate ion may not be complete. The endothermic ammonia loss from the coordination sphere, however, is covered with the reaction heat of the exothermic oxidation of a fraction of the ammonia. To clarify the nature of the decomposition intermediates, PXRD (Figure S21) and IR (Figures 8 and 9) studies were made on the decomposition intermediates prepared at 100 °C (compound **1**), 160 °C (I-160), 200 °C (I-200) and 500 °C (I-500). The temperature of the isotherm treatments was selected on the basis of the TG/DTG peak temperatures. The end-product in an inert atmosphere (I-500) is Co$_{1.5}$Mn$_{1.5}$O$_{4.2}$ ($\delta$ = 0.2) with MnCo$_2$O$_4$ (cubic spinel) structure (Figure S21 PXRD and Figure S22 far-IR). The decomposition intermediates I-160 and I-200 are amorphous products. The IR and far-IR spectra of the I-160, I-200 decomposition intermediates, and I-500 end-product show that the permanganate peaks are absent, and the intensities of the nitrate, ammonium ion, and ionic carbonate peaks appearing in I-160 decreased in the IR spectrum of I-200 and disappeared in the IR spectrum of I-500. Only the I-500 sample showed the typical vibrational modes of spinel structures related to the tetrahedral and octahedral metal-oxygen stretching bands (Figure 9) [41,42]. The average crystallite size (Scherrer method, PXRD) was found to be 16.78 nm.

The thermal decomposition reaction of compound **2** under air showed very similar features to the inert atmosphere treatments. Only an extra peak appeared at the right side of the decomposition curve of the 2nd decomposition step (Figure S18), which might be attributed to the presence of a consecutive by-reaction—probably oxidation of residual ammonia with oxygen gas. The IR (Figure S23), PXRD (Figure S24), and far-IR (Figure 9a) data of the decomposition products in air confirmed almost the same picture as in an inert atmosphere (Figure 9b).

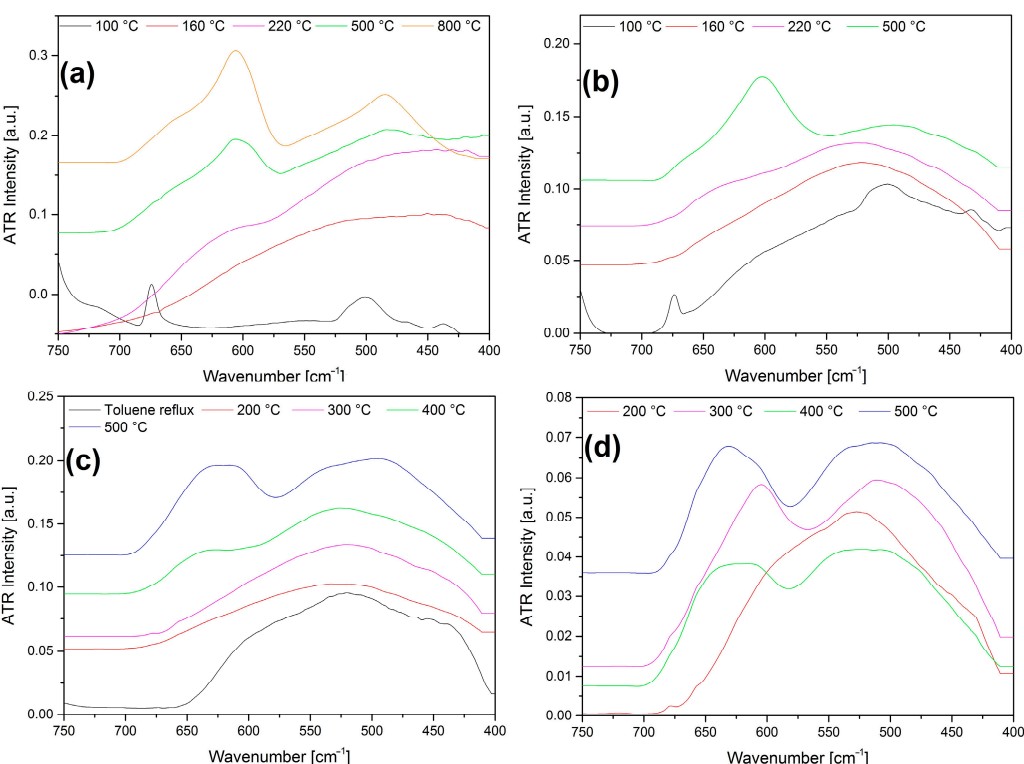

**Figure 9.** The IR spectra (between 750 and 400 cm$^{-1}$) of the thermal decomposition products of compound **2** at the indicated temperatures in (**a**) air (**b**) N$_2$, (**c**) first treated in boiling toluene, and then in air, without the water soluble part, (**d**) first treated in boiling toluene and then in air with the water soluble part.

The final decomposition product in both atmospheres is a cubic spinel-like compound with space group *Fd3m* (No. 227). The Co to Mn ratio is 1:1, and from TG data, this compound contains some oxygen surplus. The real formula can be written as Co$_{1.5}$Mn$_{1.5}$O$_{4+\delta}$. The PXRD shows that the spinel product has a MnCo$_2$O$_4$ structure. Thus, the most likely charge distribution for the participating metals can be expressed by the formula Mn$^{II/III}$[Co$^{III}_{1.5}$Mn$^{II/III}_{0.5}$O$_4$]. The thermal decomposition of [Co(NH$_3$)$_6$]Cl$_2$MnO$_4$ resulted in a spinel with the same composition (Co to Mn ratio 1:1) and structural type (MnCo$_2$O$_4$) [41]. The spinels formed from compound **2** and [Co(NH$_3$)$_6$]Cl$_2$MnO$_4$ are expected to contain 2/3 parts of manganese at the tetrahedral sites of the spinel, whereas 1/3 part of Mn, and the total cobalt content is expected to occupy the octahedral sites. In principle, the valence distribution (Co$^{II}$/Co$^{III}$ and Mn$^{II}$/Mn$^{III}$) at each site (T-4 or OC-6) may vary within a wide range, but the Mn$^{III}$ and Co$^{III}$ favor the octahedral sites. The positive $\delta$ value results in some amount of trivalent ion at the tetrahedral or tetravalent Mn on the octahedral sites to compensate for a surplus charge due to excess oxygen (metal deficit) content. Since Co$^{II}$ and Mn$^{II}$ have no preferential site within the spinel lattice [41,42,48], various (Co$^{II/III}$,Mn$^{II/III}$)[Co$^{II/III}$,Mn$^{II/III}$]O$_4$ spinels may be formed due to the Mn$^{II}$ + Co$^{III}$ = Co$^{II}$ + Mn$^{III}$ redox reactions.

To clarify the mechanism of the thermal decomposition, the evolution of the decomposition process, and to explain the differences between the reactions in an inert and oxygen-containing atmosphere, we decomposed compound **2** under toluene as a heat-absorbing medium in the presence of air. The decomposition reaction heats the inert toluene solvent until its boiling point, and the evaporation of toluene absorbs completely the liberated reaction heat [64]. In this way, the decomposition temperature is close to and cannot exceed the boiling point of toluene (110 °C). Thus, the intermediate formed at 110 °C (I-110) in the first redox reaction could be isolated.

Two series of samples were prepared, with and without removing ammonium nitrate, by washing with water before the consecutive heat treatments. First, the decomposition product was washed out with water, and the solution was left to dry. The water-soluble product was found to be orthorhombic ammonium nitrate (Figure S25). The IR and far-IR spectroscopic studies of the decomposition residue showed (Figures S26 and S27) that the primarily formed carbonate and nitrate-containing intermediates decompose on further heat treatment with the formation of a mixture of spinel structured compounds with cubic $(Co,Mn)(Co,Mn)_2O_4$ (a = 8.20 Å), and probably both types ($CoMn_2O_4$ and $MnCo_2O_4$) of crystallites were present in these samples (Figure S27). There were no cobalt-containing water-soluble components found, in contrast to the case of $[Co(NH_3)_6]Cl_2MnO_4$ and $[Co(NH_3)_5Cl](MnO_4)_2$, where the Co to Mn ratios changed in the residual material accordingly after aqueous washing. Thus, the average Co to Mn ratio 1:1 may be built up from various components of Co-rich ($MnCo_2O_4$ type) or Mn-rich ($CoMn_2O_4$) type spinels. The decomposition product was heated until 200 °C, but no crystalline phases formed. Heat treatment at 500 °C led to complete transformation into the expected spinel phase at 400 and 300 °C, with and without washing out the ammonium nitrate, respectively (Figures 9c,d and S26–S28).

The stepwise weight loss of the sample during the heating in toluene and in the consecutive heating steps until 500 °C shows that first, the crystal water and the water/ammonia formed in the redox and ligand loss reactions were released ($m$ = ~12.6%). One of the redox reaction products is ammonium nitrate. A re-oxidation of low valence metal compounds (probably manganese) was observed at 500 °C resulting in a small weight increase.

The IR spectra of the intermediates prepared by aqueous leaching showed the presence of ammonium, nitrate, and carbonate ions, and their amount decreased with increasing the heat-treatment temperature (Figures S26 and S27). Only the spinel phase was found at 500 °C, with a 5.46 nm average crystallite size. It showed that the nucleation and rearrangement into a spinel lattice were not favored, and the growth of crystallites into a spinel lattice is hindered even at 400 °C (Figure 9). It is of huge importance in catalysis when these spinels can be used even at 400 °C without crystallization-induced activity losses.

The experiments carried out without washing out of ammonium nitrate ensure a melting reaction medium for the crystallization of cobalt manganese oxides at low temperatures. The final product is the same $Mn_{1.5}Co_{1.5}O_4$ ($MnCo_2O_4$ type) spinel (Figure S28). Since the ammonium nitrate decomposes around ~200–240 °C due to the presence of catalytically active cobalt manganese oxides [41,42], the crystallization stops even below 300 °C. Accordingly, the crystallite sizes at 300 and 400 °C are practically the same (~4.0 nm). Ammonium nitrate decomposes with the formation of gases as $N_2O/H_2O$, and at 500 °C, a spinel compound formed with a 4.7 nm average crystallite size. The presence of ammonium nitrate in the decomposition process had an important influence on the properties of the cobalt manganese oxides formed due to the "solvent-like" behavior of melted ammonium nitrate, which influences properties of the cobalt manganese oxides such as valence distributions or phase relations, which have great importance in the Fischer-Tropsch catalyst systems.

The main decomposition reaction may be summarized as follows:

$$1.5[Co(NH_3)_4CO_3]MnO_4 \cdot H_2O = Mn_{1.5}Co_{1.5}O_4 + [NH_4NO_3 + 4NH_3 + 1.5CO_2 + 2H_2O].$$

The oxide ions may partially be substituted by carbonate ions ($CO_2 + O^{2-} = CO_3^{2-}$) in the intermediate compounds, and ammonium ions may also neutralize the excess charges due to the reduction of trivalent metals into divalent ones.

### 2.9. Characterization of the Solid Decomposition Products

The average crystallite sizes of $(Co,Mn)(Co,Mn)_2O_4$ spinels formed as decomposition products of compound **2** in the solid phase at 500 °C in air and under toluene with and without removal of ammonium nitrate before subsequent heating at 500 °C in air were found to be 22.03, 5.46 and 4.70 nm, respectively. The difference between the crystallinity may be attributed to the differences between the precursors formed and to the stability

of the intermediate compounds containing carbonate and ammonium ions, the stability of which influences the crystallization conditions. The melting of ammonium nitrate allows the crystallization of oxides at low temperatures into nanosized crystallites, but the decomposition of the molten ammonium nitrate prevents the further crystallization process until 400 °C and the crystal growth is resumed only at around 500 °C. The BET surface area as an important factor in catalyst design was measured; the results are given in Table 7.

**Table 7.** BET surface areas of the decomposition products formed from compound **2** under various conditions.

| Calcination Circumstances | BET Surface Area [$m^2$/g] |
|---|---|
| In air at 400 °C (2 h) | 120 |
| In air at 500 °C (2 h) | 52 |
| In air at 800 °C (2 h) | 1.8 |
| In inert atm. at 500 °C (2 h) | 10 |
| At toluene reflux temperature (2 h), washing with water and then in air at 400 °C (2 h) | 36 |
| At toluene reflux temperature (2 h), then in air at 400 °C (2 h) without washing | 37 |

The increasing temperature decreases the specific surface area of $(Co,Mn)^{T-4}(Co,Mn)^{OC-6}_2O_4$ due to sintering and crystallization processes. The SEM picture of the sample heated at 800 °C shows the formation of a well-developed crystalline material (Figure 10) with crystallite size in the μm range. Figure 10c shows the porous surface area that results in the large BET surface area (Table 7).

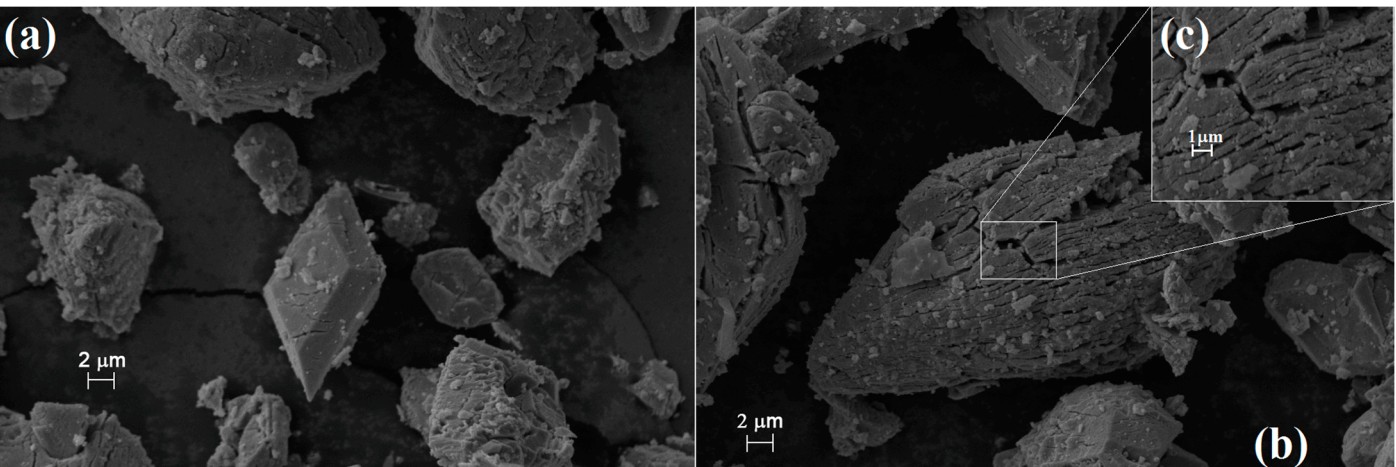

**Figure 10.** SEM picture of $Co_{1.5}Mn_{1.5}O_4$ sintered at 800 °C. Position 1 (**a**), position 2 (**b**) with 5.00 K Magnification, and Position 2 with 20.00 K magnification (**c**).

The decomposition product of formal compound **1** [4] resulted in "$CoMn_2O_4$" with a 12.4 nm average crystallite size, and its BET surface area was found to be 74.28 $m^2$/g. Compound **2** decomposed at the same temperature gave amorphous material (average crystallite size is <2 nm) and gave 120 $m^2$/g BET specific surface area.

The amorphous sample was heated at 500 °C for 2 h in air, and the sample decomposed under toluene with and without removal of the ammonium nitrate by-product was also heat treated under analog conditions. The average crystallite sizes and BET surface areas were found to be 22.03, 5.46, and 4.70nm, and the specific surface areas were 52, 26, and 37 $m^2$/g, respectively.

The final and intermediate thermal decomposition products of compound **2** (and compound **1**) are potential candidates for Fischer Tropsch synthesis, and their activity is expected to change depending on the preparation routes. The supported and unsupported cobalt manganese oxides, with and without the presence of other metal ions, are extensively studied by Fischer-Tropsch catalysts [65–74]. The quasi-intramolecular solid-phase redox reactions ensure a good possibility of adjusting the Co:Mn ratio in the formed oxides with the selection of the appropriate precursors and preparation methods [41,42]. Furthermore, the existence of a large number of isomorphic compounds among the possible precursors with other trivalent ions ($Me^{III}$ = Fe, Cr, Al) and $MO_4^-$ tetraoxometallate anions ($ClO_4^-$, $ReO_4^-$, $1/2SO_4^{2-}$, $1/2CrO_4^{2-}$). The $[M^{III}L_n](MnO_4,ClO_4)(YO_4, Cr_2O_7, S_2O_8)$ mixed salts (where Y = S, Cr, etc.) [36,38,75–86] gives enormous possibilities for adjusting the compositions and performances of new catalysts. The low cobalt and permanganate-containing mixed crystals with "inert" cations (e.g., Al) and gas-forming anions (sulfate, perchlorate) ensure the in situ preparation of "supported" (e.g., with the co-formation of $Al_2O_3$ support) Co-Mn oxide catalysts.

Therefore, the detailed catalytic properties of these samples synthesized in various reaction routes and from various amminecobalt(III) permanganate precursors with various Co to Mn ratios (1:1–1:3) (including the catalysts prepared from the title compound of this paper) will be compared in our forthcoming publications.

## 3. Materials and Methods

Ammonia (25% aq. solution), ammonium carbonate, cobalt(II) nitrate hexahydrate, potassium permanganate, barium hydroxide, sodium sulfate decahydrate, aluminum sulfate-18hydrate, ethanol, benzene, diethyl ether, chloroform, toluene, $D_2O$ and other analytical reagents were supplied by Deuton-X Ltd. (Érd, Hungary). Sodium permanganate was prepared using methods known as barium manganate and sodium sulfate [50].

### 3.1. Preparation of [Co(NH₃)₄CO₃]MnO₄.H₂O (Compound **2**)

2.58 g (0.01 mol) $[Co(NH_3)_4CO_3]NO_3$, prepared according to Mansouri [4], was dissolved in 120 mL distilled water, and to this solution, 2.56 mL $NaMnO_4$ (40 w% solutions) (0.01 mol) was suddenly added while stirring. Immediately, dark purple, shiny micro-sized crystals were formed, which were filtrated with a G4 pore size glass filter, and the crystals were washed with 50 mL 0 °C cold distilled water. The weight of the final product was 2.8634 g (88.4%). The $[Co(NH_3)_4CO_3]MnO_4.H_2O$ formed is slightly soluble in water (0.16 g/L), and the pH of the solution is 5.33. It is not soluble in ethanol and decomposes in diethyl ether and chloroform. Plate-like single crystals of compound **2** with 0.1 mm × 0.1 mm × 0.03 mm dimensions were selected for SXRD measurements.

### 3.2. Preparation of [Co(ND₃)₄CO₃]MnO₄.D₂O (Compound **2-D**)

Compound **2-D** was prepared with the dissolution of compound **1** in $D_2O$ with a subsequent crystallization over freshly prepared CaO in a desiccator. The cobalt and manganese content of compounds **1** and **2** were determined by atomic emission spectroscopy with a Spectro Genesis ICP-OES (SPECTRO Analytical Instruments GmbH, Kleve, Germany) instrument. The calibration was conducted with a Merck multielement standard (Merck Chemicals GmbH, Darmstadt, Germany). The ammonia content was determined by gravimetry as $(NH_4)_2PtCl_6$, as has been given in detail earlier [42,43].

### 3.3. Single Crystal X-ray Diffraction

The SXRD measurements were performed on a Rigaku XTALab Synergy-R diffractometer (Tokyo, Japan) using CuKα radiation at 100 K. Crystal data and details of the structure determination and refinement are listed in Tables 2 and S1. CrysAlisPro v 1.171.42.58a was used for data collection and data reduction. Olex2 (version 1.5) [87] and Mercury (version 2020.2.0) [88] were used for molecular graphics and analyzing crystal packing. Data were collected at 100 K using 56 ω scans. A total of 8890 frames were collected (0.5° rotation,

0.05 s and 0.1 s exposure time/frame for the low and high θ regions, respectively). The structure was solved using the intrinsic phasing method as implemented in SHELXT (version 2018/3) [89]. Refinement was carried out using Olex2 (version 1.5) [87] and SHELXL (version 2018/3) [90] with the full matrix least squares method on $F^2$. All non-hydrogen atoms were refined anisotropically. Hydrogen atoms were found in different Fourier maps, and their positions were refined using the riding model. Analytical absorption correction was applied to the data.

### 3.4. Hirshfeld Surface Analysis

The Hirshfeld surface analysis was performed using CrystalExplorer (version 21.3) [91]. Hydrogen bond analysis was performed using Platon software (version 2023.1) [92].

### 3.5. Powder X-ray Diffraction

The PXRD patterns of compounds **1** and **2** were acquired with the use of a Philips Bragg-Brentano para focusing goniometer (Amsterdam, The Netherland), copper Kα radiation, 1.5406/1.5444 Å) in the 2θ range of 4–70°, (step size was 0.02° with 1 s interval time). The other details of the XRD analysis are given in [57].

### 3.6. Specific Surface Area Measurements

The specific surface areas of intermediates and final decomposition products of compounds **1** and **2** were determined with the use of the Brunauer-Emmett-Teller (BET) equation ($N_2$ gas adsorption data was collected at −196 °C) and an Autosorb 1C (Quantachrome, Boynton Beach, FL, USA) instrument. The samples were evacuated before the measurements at 100 °C for 24 h. The details of measurement have been given earlier [57].

### 3.7. Vibrational Spectroscopy

The IR and far-IR spectra of compounds **1**, **2**, and **2**-D were recorded in an Attenuated Total Reflectance (ATR) mode with 16 scans (acquired with a resolution of 4 cm$^{-1}$) on a Bruker Alpha FT-IR (Bruker, Ettingen, Germany) and a Biorad-Digilab FTS-30 instrument (Biorad, Budapest, Hungary), respectively. The details of measurements were given in [57].

The Raman spectra of compounds **1** and **2** were recorded between 2000 and 100 cm$^{-1}$ on a Horiba Jobin-Yvon LabRAM microspectrometerat (Kyoto, Japan) 298 and 123 K with an external 785 nm diode laser source (~80 mW), which was coupled to an optical microscope (Olympus BX-40). The laser beam was focused on an objective of 20×. To avoid the degradation of the studied compounds, a D2 intensity filter was used to decrease the laser power to 1%. The confocal hole (1000 μm) and a monochromator (950 grooves mm$^{-1}$ grating) were used for light dispersion. The resolution was 4 cm$^{-1}$. The exposure times were selected to 120 s to produce intensive peaks. The low-temperature Raman measurements were conducted at 123 K (−150 °C) with the use of a Linkam THMS600 temperature control stage cooled by liquid $N_2$. The details of the measurement are given in [57].

### 3.8. UV-Vis Spectroscopy

The UV-VIS diffuse reflectance spectra of compounds **1** and **2** were recorded at room temperature with the use of a Jasco V-670 UV–VIS instrument (NV-470 integrating sphere, $BaSO_4$ standard).

### 3.9. Scanning Electron Microscopy

SEM pictures of a decomposition product of compound **2** were recorded on a JEOL JSM-5500LV scanning electron microscope. The sample specimen was fixed on sample holders (Cu/Zn alloy, carbon tape) and sputtered with a conductive Au/Pd layer for imaging.

### 3.10. DSC Studies

The DSC measurements were conducted on a Perkin Elmer DSC 7 instrument between $-140$ and 25 °C and 25 and 400 °C in an unsealed aluminum pan; the sample masses were between 3 and 5 mg, and a heating rate was selected to be 5 °C/min under a continuous $N_2$ or $O_2$ flow (20 cm$^3$ min$^{-1}$).

### 3.11. Thermal Studies

The TG-DTG-DTA/TG-MS data were collected with the use of a modified TGS-2 thermobalance (Perkin Elmer, Shelton, CT, USA), which was coupled to a HiQuad quadrupole mass spectrometer (Pfeiffer Vacuum, Bruehl, Germany). A small amount of samples (~1 mg) were used due to explosion sensitivity, and a Pt sample pan was used. The thermal decomposition was followed from ambient temperature to 600 °C at a 10 °C min$^{-1}$ heating rate in argon or air carrier gases (flow rate = 140 cm$^3$ min$^{-1}$). Significant ions were monitored between $m/z$ of 2 and 88 in a SIM (Selected Ion Monitoring) mode.

### 4. Conclusions

An easy reaction route was developed to prepare phase-pure $[Co(NH_3)_4CO_3]MnO_4$ (compound **1**) and its hydrated form, $[Co(NH_3)_4CO_3]MnO_4 \cdot H_2O$ (compound **2**). Compound **1** is triclinic, whereas compound **2** is orthorhombic. The lattice parameters were determined by PXRD and SXRD methods for compounds **1** and **2**, respectively. The detailed structure of compound **2** was determined, and the role of hydrogen bonds in the structural motifs was clarified.

Compounds **1** and **2** were studied by IR, far-IR, as well as by room temperature and low temperature (123 K) Raman studies, and the vibrational modes of the cis—$CoO_2N_4$ core, ammonia, the carbonate ligands, the permanganate ions, and the crystal water were assigned. UV spectroscopic studies show the distortion of the octahedral geometry due to dehydration, resulting in the loss of a part of the hydrogen bonds mediated by the crystal water and the complex cation.

The thermal decomposition of compound **2** (its decomposition intermediate is compound **1**) was tracked by DSC and TG-MS methods. The dehydration can be completed without the elimination of ammonia. The thermal decomposition of compound **2** is a solid phase quasi-intramolecular redox reaction between the ammonia ligands and the permanganate anions. Gaseous ammonia and the oxidation products of ammonia ($H_2O$, NO, $N_2O$, and $CO_2$) form together with amorphous cobalt manganese oxides containing ammonium and carbonate (as well as nitrate) anions. The thermal decomposition of compound **2** under toluene at 110 °C showed that one of the decomposition intermediates is ammonium nitrate. The heat treatment of the primary decomposition products resulted in the formation of $Co_{1.5}Mn_{1.5}O_4$ spinel with $MnCo_2O_4$ structure. The solid compound **1** gave the spinel at 500 °C both in inert and air atmosphere. The thermal decomposition intermediate prepared at 110 °C in toluene with and without removal of the ammonium nitrate by aqueous leaching gave the spinel already at 400 and 300 °C, respectively. The molten $NH_4NO_3$ is a medium that starts the crystallization of the spinel, but its decomposition stops the crystal growth. Therefore, the particle size of the spinel in these samples prepared at 300 and 400 °C could be reduced to ~4.0 nm and to 5.7 nm at 500 °C.

The nano-sized mixed cobalt manganese oxides are potential candidates as Fischer-Tropsch catalysts. Such studies are in progress together with the preparation of other mixed Co-Mn oxides from other permanganate salts of various ammine cobalt complex ions (e.g., $[Co(NH_3)_6]X_2MnO_4$, (X = Cl, Br), $[Co(NH_3)_5Cl](MnO_4)_2$ and $[Co(NH_3)_6](MnO_4)_3$ with Co to Mn ratio from 1:1 to 1:3.

**Supplementary Materials:** The following supporting information can be downloaded at: https://www.mdpi.com/article/10.3390/inorganics12040094/s1, Figure S1: Powder-XRD study of the reaction product of synthesis root described in ref. [4]; Figure S2: Analytical range IR study of heat treatment residues of compound **2**; Figure S3: Powder-XRD of Compound **2**, partially (2 h), and

completely dehydrated products (4 h)—Compound **1**; Figure S4: Powder-XRD of (**a**) Compound **1** and (**b**) $[Co(NH_3)_4CO_3]ClO_4$; Figure S5: Measured and theoretical Powder-XRD of Compound **2**; Figure S6: Crystal packing in compound **2**. View of the unit cell along the *a*, *b* and *c* axes; Figure S7: Correlation analysis diagram (**a**) internal and (**b**) external modes for ammonia ligands in compound **2**; Figure S8: Raman spectra of compound **1** measured with 785 nm laser; Figure S9: Raman spectra of compound **2** measured with 785 nm laser; Figure S10: IR spectra of (**a**) compound **1**, (**b**) compound **2**, and (**c**) compound **2-D**; Figure S11: IR spectra of compound **1**, compound **2** and partially deuterated compound **2** (**2-D**) between 1800 and 400 cm$^{-1}$; Figure S12: Correlation analysis diagram (**a**) internal and (**b**) external modes for carbonate ligand in compound **2**; Figure S13: The far-IR spectra of compound **2**; Figure S14: Correlation analysis diagram for central Co$^{III}$ ion in compound **2**; Figure S15: The Raman spectra of compounds **1** and **2** recorded at 123 K in the far-IR region; Figure S16: Correlation analysis diagram for the permanganate ion in compound **2**; Figure S17: UV-VIS spectra of compounds **1** and **2**; Figure S18: TG (green), DTG (blue), and DTA (red) curve of compound **2** in an inert atmosphere; Figure S19: TG (green), DTG (blue), and DTA (red) curve of compound **2** in air; Figure S20: DSC of compound **2** under in an (**a**) $O_2$ and (**b**) $N_2$ atmosphere; Figure S21: PXRD of the thermal decomposition products of compound **2** under an inert atmosphere (I-100 (compound **1**), I-160, I-200 and I-500); Figure S22: Far-IR spectra of the thermal decomposition products of compound **2** under an inert atmosphere; Figure S23: Far-IR spectra of the thermal decomposition products of compound **2** under an aerial atmosphere; Figure S24: PXRD of the thermal decomposition products of compound **2** under an aerial atmosphere; Figure S25: PXRD of water-soluble product of I-110 °C; Figure S26: IR spectra of the thermal decomposition products of compound **2** under boiling toluene and after under air. Heat treatment (**a**) without and (**b**) with water-soluble part (left back after decomposition in toluene); Figure S27: Far-range IR spectra of the thermal decomposition products of compound **2** under boiling toluene and after under air. Heat treatment (**a**) without and (**b**) with water-soluble part (left back after decomposition in toluene); Figure S28: PXRD of the thermal decomposition products of compound **2** under boiling toluene and after under air. Heat treatment (**a**) without and (**b**) with water-soluble part (left back after decomposition in toluene). Table S1: Crystallographic parameters of the orthorhombic compound **2**; Table S2: Bond lengths [Å] and angles [°] in the structure of compound **2**; Table S3: Atomic coordinates ($\times 10^4$) and equivalent isotropic displacement parameters (Å$^2 \times 10^3$) in compound **2**. U(eq) is defined as one-third of the trace of the orthogonalized U$^{ij}$ tensor; Table S4: Anisotropic displacement parameters (Å$^2 \times 10^3$ in compound **2** The anisotropic displacement factor exponent takes the form: $-2\pi^2[h^2 a \cdot 2U^{11} + \ldots + 2hka \cdot b \cdot U^{12}]$; Table S5: Hydrogen coordinates ($\times 10^4$) and isotropic displacement parameters (Å$^2 \times 10^3$) in compound **2**; Table S6: Torsion angles [°] in compound **2**; Table S7: The internal vibrational modes of $CoN_4O_2$ core and their tentative assignments in the IR and Raman spectra of compounds **1** and **2**. (C$_{2v}$ symmetry is assumed); Table S8: Assignment of the permanganate vibrational modes in the IR and Raman spectra of compounds **1** and **2**.

**Author Contributions:** Conceptualization, L.K. and K.A.B.; methodology, L.K.; formal analysis, V.M.P. and Z.H.; investigation, K.A.B., Z.D., L.B., B.B.H. and A.F.; writing—original draft preparation, L.K.; writing—review and editing, Z.H., K.A.B. and V.M.P.; visualization, supervision, Z.H. and K.A.B.; All authors have read and agreed to the published version of the manuscript.

**Funding:** This research was funded by ÚNKP-21-3, ÚNKP-22-3, and ÚNKP-23-3 New National Excellence Program of the Ministry for Culture and Innovation from the Source of The National Research, Development and Innovation found (K.A.B.) and by the European Union and the State of Hungary, co-financed by the European Regional Development Fund, grant number VEKOP-2.3.2-16-2017-00013 (L.K.) and VEKOP-2.3.3.-15-2017-00018 (Z.D.).

**Conflicts of Interest:** László Kótai was employed by the Deuton-X Ltd. company. The Deuton-X Ltd. company was not involved in the study design, collection, analysis, interpretation of data, the writing of this article or the decision to submit it for publication. The remaining authors of the paper declare that the research was conducted in the absence of any commercial or financial relationships that could be construed as a potential conflict of interest.

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
