# Peer review of "Insight into the Structure and Redox Chemistry of [Carbonatotetraamminecobalt(III)] Permanganate and Its Monohydrate as Co-Mn-Oxide Catalyst Precursors of the Fischer-Tropsch Synthesis"

_inorganics, doi:10.3390/inorganics12040094_

Round 1

Reviewer 1 Report

Comments and Suggestions for Authors

This is a comprehensive and well-analyzed experimental study that produced and characterized [carbonatotetraamminecobalt(III)] permanganate, its monohydrate, and corresponding perchlorate samples.  The work and analysis appear to be of high quality, which could be published in its present form, except there are a couple of unusual symbols (noted below) that appear to be font rendering errors that should be corrected prior to publication.  Therefore, I recommend publication after minor revision, in consideration of the following issues.

Line 484 contains a character with which I am unfamiliar.  It appears to be a bird with an asterisk near its tail.  I suppose that this is an incorrect electronic rendering of another symbol that the authors intended, but if not, please provide a description of what this symbol represents.  The same bird symbol appears in lines 495, 558, and 559.  Is this supposed to be pi?

Similarly, in line 698, a vortex or swirl symbol appears in front of “m”.  From the context, this apparently should be the Greek letter “mu” to stand for “micron” or “micrometer”.

Minor grammatical/typographical issues:

Lines 41 and 584 contain degree symbols that do not look like those in the rest of the document.  (They appear to be superscripted letters “o” and not true degree symbols, as are used in the rest of the manuscript.)  They should be consistent throughout.

In line 69, “is” should be “are”.

In line 156, “salt” should be “salts”.

In line 642, there is a floating comma that makes appear as if a word or phrase is missing.  However, the writing appears to make sense, so please remove the extraneous space before the comma.

Comments on the Quality of English Language

The English quality is generally fine.  It could use some minor polishing, but the authors' intended meaning can be discerned.

Author Response

Reviewer #1:

  1. Line 484 contains a character with which I am unfamiliar. It appears to be a bird with an asterisk near its tail. I suppose that this is an incorrect electronic rendering of another symbol that the authors intended, but if not, please provide a description of what this symbol represents.  The same bird symbol appears in lines 495, 558, and 559.  Is this supposed to be pi?.

    We are grateful for the Reviewer’s comment. We have corrected the symbols to be printed as the corresponding greek letters.
  2. Similarly, in line 698, a vortex or swirl symbol appears in front of “m”. From the context, this apparently should be the Greek letter “mu” to stand for “micron” or “micrometer”.

Thank you very much, we have corrected the mistake.

  1. Lines 41 and 584 contain degree symbols that do not look like those in the rest of the document. (They appear to be superscripted letters “o” and not true degree symbols, as are used in the rest of the manuscript.) They should be consistent throughout.

We are grateful for the Reviewer’s comment. We have corrected all the degree symbols to the correct ones.

  1. In line 69, “is” should be “are”. In line 156, “salt” should be “salts”. In line 642, there is a floating comma that makes appear as if a word or phrase is missing. However, the writing appears to make sense, so please remove the extraneous space before the comma.

We are grateful to the Reviewer for helping us to find grammatical and style mistakes and improve our paper in this way. We have checked the manuscript for typographic errors and improved the grammar.

Reviewer 2 Report

Comments and Suggestions for Authors

Journal Inorganics (ISSN 2304-6740)

Comments on Manuscript inorganics-2911479

Title : Insight into the structure and redox chemistry of [carbonatotetraamminecobalt(III)] permanganate and its monohydrate as Co-Mn-oxide catalyst precursors of the Fischer-Tropsch Synthesis.

This work presents the syntheses of mixed CoMn oxide, proposing two different methods, which were compared to a previous work, using “ [Co(NH3)4CO3]NO3 and NaMnO4 in aqueous solution. Its thermal dehydration at 100 °C resulted in phase-pure [Co(NH3)4CO3]MnO4, named  (compound 1) and Compounds 1 and 2, which were treated thermally and calcined at different temperatures. They were fully characterized aiming to evaluate in the Fischer-Tropsch synthesis.

In fact, the compounds 1 and 2 presented different structure as  shown in table 2 presenting different cell  parameters and compared with a reference catalyst. The Fig.8 shows that the sample calcined in air is stable after calcination of 500C, and therefore, after calcination at 800C they obtained the mixed oxide. As observed in Table 7 , when calcined at 800C the material was sintered and the BET area decayed form 120  after to 1.8 m2/g, which seems that all pores were blocked and the dispersion of Co and Mn may not easily reduced.. The question is what reduction temperature is necessary to disperse the metallic Co and Mn oxide, and how the Mn oxide act to disperse the surface-active sites.  The FTS requires active site.  Therefore. the mixed oxide calcined at 800C must be reduced and well dispersed.

The TGA results in Fig.7 shows the formation of high amounts at 200 C when treated with an inert gas and above 300C increasing release of water when treated with air. This is unclear and please explain, because the Nitrogen compounds are easily eliminated at this temperature.

The main objective is to synthesize new non supported catalyst. There are many reports for FTS with mixed oxides, which were not critically presented in this work, for comparison.

Author Response

We are grateful for the Reviewer’s comment and question. We tried to completely fulfil the requests. The answers/explanations and changes are as follows.

  1. In fact, the compounds 1 and 2 presented different structure as shown in table 2 presenting different cell parameters and compared with a reference catalyst.

The change of the cell parameters is due to the crystal water content, which changes the symmetry of the lattice. The water molecules fill the voids; without their presence, the cell of the hydrate rearranges and transforms into a triclinic crystal system of lower-symmetry. We have added this comment to the text.

  1. The Fig.8 shows that the sample calcined in air is stable after calcination of 500C, and therefore, after calcination at 800C they obtained the mixed oxide. As observed in Table 7, when calcined at 800C the material was sintered and the BET area decayed form 120 after to 1.8 m2/g, which seems that all pores were blocked and the dispersion of Co and Mn may not easily reduced. The question is what reduction temperature is necessary to disperse the metallic Co and Mn oxide, and how the Mn oxide act to disperse the surface-active sites. The FTS requires active site. Therefore, the mixed oxide calcined at 800C must be reduced and well dispersed.

The decreasing BET surface area (due to the increase in temperature) might be due to more than one reason. During the heat treatment, vacancies are formed in the structure of the oxide products, especially in an inert atmosphere, which results in broadened peaks of the powder X-ray diffractograms (see ESI Figure S24). The porous structure is formed with highly random Mn-Co dispersion. Therefore, in this way high BET surface area can be reached.

At higher temperatures (800 °C), especially in an oxygen-rich atmosphere, these structural vacancies partially disappear due to the building of oxygen atoms into the structure. Together with the shrinking of the material (due to the heating), this makes it possible to form a well-crystallized structure (see ESI Figure S24). This means a sintered material, resulting in a lower BET surface area. Furthermore, during this transformation process, the dispersion of the Mn and Co can easily decrease. Thus, directly not only the temperature is responsible for the decrease of the dispersion and the chemical reduction, but rather the structural change is the cause. These phenomena will be studied for this and other cobalt manganese oxides in our forthcoming paper dealing with surface chemistry and catalytic properties of the materials prepared in the given reaction route. Again, thanks to our reviewer for calling our attention to the importance of this question. 

  1. The TGA results in Fig.7 shows the formation of high amounts at 200 C when treated with an inert gas and above 300C increasing release of water when treated with air. This is unclear and please explain, because the Nitrogen compounds are easily eliminated at this temperature.

We thank the Reviewer for the professional question. After reviewing Figure 7, it turned out, that the (a) and (b) notions of the figure were mixed and the right form would be:

Figure 7. TG-MS curves of m/z=18, 17, 16, 30 and 44 ions during the thermal decomposition of compound 2 in air (a) and inert (b) atmosphere.”

According to the new figure caption, the difference makes sense. In air, during the heat treatment, there is an excess amount of oxygen, and together with the temperature it easily decomposes the NH3 into NOx, and H2O, at a relatively low temperature. On the other hand, in the case of inert atmosphere heat treatment, when only the oxygen atoms of the CO32- and MnO4- are available,  4 mol NH3 and 7 mol ‘O’ atoms show strong oxygen deficiency to oxidize the ammonia molecules completely. In this way,  the temperature causes the decomposition (pyrolysis) of the non-oxidized residual NH3, which requires higher temperature.

  1. The main objective is to synthesize new non supported catalyst. There are many reports for FTS with mixed oxides, which were not critically presented in this work, for comparison.

We are grateful for the Reviewer’s comment. In this paper, we mentioned that there were already done some FTS by Mansouri et al. [4,6] with a Co-Mn-oxide prepared from the title compound.  However, our results showed that they have used contaminated catalysts. According to the reviewer’s request, we introduced ca. 10 citations to show the activity of these kinds of Co-Mn oxides in FTS and give an explanation to point to the future directions of our investigations in this field. 

Reviewer 3 Report

Comments and Suggestions for Authors

Comments to Kótai et al.

Summary

The study deals with the structure and redox reactions of carbonatotetraamminecobalt(III) permanganate and its hydrate. More specifically, the authors synthesize the compounds and employ analytical tools like Raman spectroscopy and X-ray diffraction to determine the structure of the compounds and the reaction products. Furthermore, the study considers some decomposition products as potential candidates for Fischer-Tropsch catalysts.

General comments

This is an extensive and careful work on a topic well suited for the Inorganics journal. Moreover, the structure of the manuscript is adequate for a scientific journal and the illustrations and tables are of good quality. Also, the English language is fine. As for the scientific content, it is solid and of potential interest. The description of the results from some of the analytic methods could have additional references for complicated concepts.

Specific comments

Lines 28-29: It would be good to write out also the full words at the first use of acronyms (or in this case maybe rather initialisms).

Lines 462-464: This is a bit difficult to follow. What are rho, omega and tau? Would it be possible to provide a lucid geometric scheme?

Line 561: show without singular s (position of ion and lack of stretching mode bands)

Lines 569-570: The unnecessary length and complication of the sentence make its message ambivalent. One problem is that the coordinating conjunction or should combine two parts of equal syntactic importance, which in the sentence under scrutiny would combine the last main clause (starting with this role) with the clause starting with thus instead of with the subordinate if clause, which probably was the intention. Moreover, the word thus would indicate that the last part of the sentence is a conclusion following from the first part, when in fact the last part is an argument supporting the first part hypothesis.

Lines 671-674: Perhaps you could elaborate on why the presence of ammonium nitrate potentially is favorable for Fischer-Tropsch catalysis.

Lines 842-843: Grammatically, the last sentence could be more disciplined. Now you have the study of a chemical together with chemicals themselves.

Author Response

  1. The description of the results from some of the analytic methods could have additional references for complicated concepts.

We are grateful for the Reviewer’s comment and suggestion. We have added the required references to the experimental section.

  1. Lines 28-29: It would be good to write out also the full words at the first use of acronyms (or in this case maybe rather initialisms).

We are grateful for the Reviewer’s suggestion. We have modified the abstract.  

  1. Lines 462-464: This is a bit difficult to follow. What are rho, omega and tau? Would it be possible to provide a lucid geometric scheme.

We are grateful for the Reviewer’s comment and suggestion. These symbols are notations of the molecular vibrational modes. We are sending a figure in our response, that shows the meaning of the given Greek letters in visual form.  

  1. Line 561: show without singular s (position of ion and lack of stretching mode bands).

Many thanks to  the Reviewer’s comment. We have modified the text.

  1. Lines 569-570: The unnecessary length and complication of the sentence make its message ambivalent. One problem is that the coordinating conjunction or should combine two parts of equal syntactic importance, which in the sentence under scrutiny would combine the last main clause (starting with this role) with the clause starting with thus instead of with the subordinate if clause, which probably was the intention. Moreover, the word thus would indicate that the last part of the sentence is a conclusion following from the first part, when in fact the last part is an argument supporting the first part hypothesis

We are grateful for the Reviewer’s suggestion to change this part of the manuscript. We have rephrased the above-mentioned section.

  1. Lines 671-674: Perhaps you could elaborate on why the presence of ammonium nitrate potentially is favorable for Fischer-Tropsch catalysis.

We express our thanks for the Reviewer’s comment. We have added other details to this section of our manuscript.

  1. Lines 842-843: Grammatically, the last sentence could be more disciplined. Now you have the study of a chemical together with chemicals themselves.

We are grateful for the Reviewer’s suggestion, and we have rewritten the sentence.

Round 2

Reviewer 2 Report

Comments and Suggestions for Authors

Authors revised the manuscript and responses and both were very well done. I recommend publication of this manuscript in the present form.

Author Response

Thank you for your recommendation.